# Bottom-up evolution of perovskite clusters into high-activity rhodium nanoparticles toward alkaline hydrogen evolution

Gaoxin Lin[1,2,11], Zhuang Zhang[1,2,11], Qiangjian Ju[1,2,11], Tong Wu[1,2], Carlo U. Segre [3], Wei Chen [4], Hongru Peng[5], Hui Zhang [6], Qiunan Liu[7], Zhi Liu [5,6], Yifan Zhang[1,2], Shuyi Kong [1,2], Yuanlv Mao[1,2], Wei Zhao[1,2], Kazu Suenaga[7], Fuqiang Huang [1,8] ✉ & Jiacheng Wang [1,2,9,10] ✉

Self-reconstruction has been considered an efficient means to prepare efficient electrocatalysts in various energy transformation process for bond activation and breaking. However, developing nano-sized electrocatalysts through complete in-situ reconstruction with improved activity remains challenging. Herein, we report a bottom-up evolution route of electrochemically reducing $Cs_3Rh_2I_9$ halide-perovskite clusters on N-doped carbon to prepare ultrafine Rh nanoparticles (~2.2 nm) with large lattice spacings and grain boundaries. Various in-situ and ex-situ characterizations including electrochemical quartz crystal microbalance experiments elucidate the Cs and I extraction and Rh reduction during the electrochemical reduction. These Rh nanoparticles from $Cs_3Rh_2I_9$ clusters show significantly enhanced mass and area activity toward hydrogen evolution reaction in both alkaline and chlor-alkali electrolyte, superior to liquid-reduced Rh nanoparticles as well as bulk $Cs_3Rh_2I_9$-derived Rh via top-down electro-reduction transformation. Theoretical calculations demonstrate water activation could be boosted on $Cs_3Rh_2I_9$ clusters-derived Rh nanoparticles enriched with multiply sites, thus smoothing alkaline hydrogen evolution.

Hydrogen, as an energy carrier, is critical to utilize the renewable energy including wind, solar and hydropower for sustainable development[1-6]. The alkaline hydrogen evolution reaction (HER) with sluggish kinetics limits the efficiency of $H_2$ generation in water electrolysis and chlor-alkali electrolysis due to the additional step of water dissociation[7-12]. It starts from the cleaving of the H−OH bond coupled with an electron to form the adsorbed hydrogen atom ($H_{ad}$, Volmer step). Then, $H_2$ is produced by the combination of two

[1]State Key Lab of High Performance Ceramics and Superfine microstructure, Shanghai Institute of Ceramics, Chinese Academy of Sciences, 201899 Shanghai, China. [2]Center of Materials Science and Optoelectronics Engineering, University of Chinese Academy of Sciences, 100049 Beijing, China. [3]Department of Physics & Center for Synchrotron Radiation Research and Instrumentation, Illinois Institute of Technology, Chicago, IL 60616, USA. [4]Department of Mechanical, Materials and Aerospace Engineering, Illinois Institute of Technology, Chicago, IL 60616, USA. [5]School of Physical Science and Technology, ShanghaiTech University, 201210 Shanghai, China. [6]State Key Laboratory of Functional Materials for Informatics, Shanghai Institute of Microsystem and Information Technology, Chinese Academy of Sciences, 200050 Shanghai, China. [7]SANKEN, Osaka University, Ibaraki 567-0047, Japan. [8]State Key Laboratory of Rare Earth Materials Chemistry and Applications, College of Chemistry and Molecular Engineering, Peking University, 100871 Beijing, China. [9]Hebei Provincial Key Laboratory of Inorganic Nonmetallic Materials, College of Materials Science and Engineering, North China University of Science and Technology, 063210 Tangshan, China. [10]School of Materials Science and Engineering, Taizhou University, 318000 Taizhou, Zhejiang, China. [11]These authors contributed equally: Gaoxin Lin, Zhuang Zhang, Qiangjian Ju. ✉e-mail: huangfq@mail.sic.ac.cn; jiacheng.wang@mail.sic.ac.cn

$H_{ad}$ (Tafel step) or the interaction of $H_{ad}$ and a water molecule (Heyrovsky step)[7,13–15]. Thus, the properties of water dissociation, and hydroxy radical and hydrogen adsorption are crucial for alkaline HER.

To achieve high activity toward these basic steps in alkaline HER, it is necessary to design a functional composite with multiple active centers to facilitate the overall reaction[16–21]. However, such a comprehensive system not only brings difficulty in material synthesis, but also is harmful to the durability of the catalyst due to element or phase segregation[22–26], especially in a harsh chlor-alkali solution. The potential-driven structural reconstruction of a pre-catalyst during the working conditions is an efficient method to enhance the electrochemical performance[26–29]. The reconstruction of pre-catalysts could obtain the amorphous or defect-rich structure, increased accessible surface area, optimized adsorption properties, and promoted charge transfer[30–33], thus leading to the improvement of activity and stability. For example, $Ni_2P$ could be transformed in situ into $Ni_2P/NiO_x$ core-shell structure during oxygen evolution reaction[34]. It could accelerate water adsorption and dissociation kinetics due to unique heterostructures and multiple active centers. Besides, the $SrIrO_3$ perovskite electrode experiences self-reconstruction during the alkaline HER because of lattice $Sr^{2+}$ leaching[35]. The formed metallic Ir on the surface of perovskite results in the remarkable activity enhancement as well as excellent stability. However, most pre-catalysts undergo the surface-reconstruction, which leads to the low component utilization as the inert internal part is inaccessible to the surface catalysis. Moreover, the reconstruction degree is highly relevant to the reaction environment, and it may be changed with pH, temperature, electrolyte, and applied potential[31,36–38], which is disadvantageous to the industrial extreme condition. Therefore, the electrocatalysts consisting of

single-component nanoparticles prepared via complete in situ reconstruction are highly desired to avoid the above problems during the alkaline HER.

Herein, we develop a bottom-up evolution route to prepare high-activity Rh nanoparticles via in situ electrochemical reduction of new $Cs_3Rh_2I_9$ perovskite clusters. These electrochemically reduced Rh nanoparticles could be used as a highly efficient HER catalyst in both alkaline and chlor-alkali electrolyte. The new halide-perovskite compound $Cs_3Rh_2I_9$ with dimer unit $[Rh_2I_9]^{3-}$ separated by Cs ions as the pre-catalyst of electrochemically synthesized Rh nanoparticles was synthesized by solid state reaction (Fig. 1a). It could be dissolved in N, N-dimethylformamide (DMF). The unique zero-dimensional structure allows it to be downsized into small clusters on a polar nitrogen-doped carbon (NC) support by a simple dissolution-precipitation method (Fig. 1b). The high surface energy of $Cs_3Rh_2I_9$ clusters on NC ($Cs_3Rh_2I_9/NC$) could promote an electrochemical self-reduction and bottom-up evolution, leading to the formation of unique Rh nanoparticles with larger lattice spacings and lower atomic coordination number (Fig. 1b). In sharp contrast, such Rh nanoparticles cannot be formed from bulk $Cs_3Rh_2I_9$ and liquid reduction of $RhCl_3$ by $NaBH_4$ (Fig. 1c, d). The complete reconstructed $Cs_3Rh_2I_9/NC-R$ could significantly reduce the barrier of water dissociation in alkaline HER. Therefore, $Cs_3Rh_2I_9/NC-R$ exhibits high mass activity of 772.1 mA mg$^{-1}_{Rh}$ in a chlorine-alkali electrolyte, which is about 2.5 times and 35.5 times that of liquid-reduced Rh/NC with similar particle size and electrochemically reduced $Cs_3Rh_2I_9-R$ with the larger size, respectively (Fig. 1e, f). And it also shows the negligible activity loss after 50 h durable measurement.

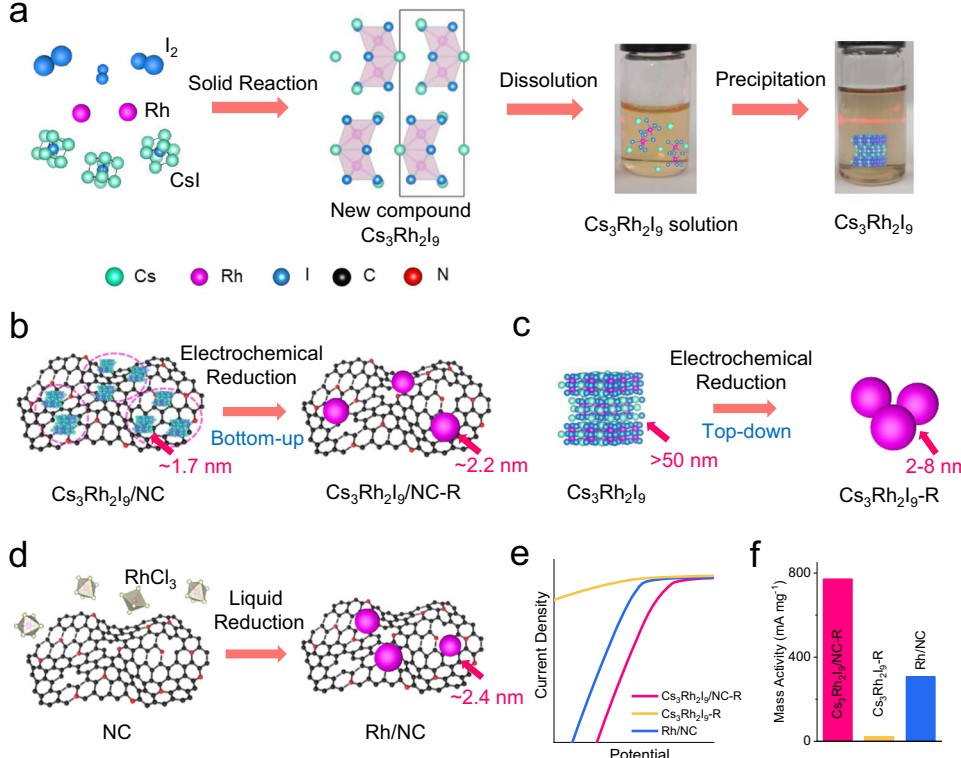

**Fig. 1 | Schematic illustration of synthesizing perovskite $Cs_3Rh_2I_9$ bulk crystals and clusters $Cs_3Rh_2I_9/NC$, and their electrochemical reduction into metallic Rh particles toward alkaline HER. a** Synthesis of $Cs_3Rh_2I_9$ bulk crystal via a solid reaction using $I_2$, Rh, and CsI at 800 °C. The $Cs_3Rh_2I_9$ crystal could demonstrate a dissolution-precipitation phenomenon in N, N-dimethylformamide (DMF). **b** Electrochemical reduction of $Cs_3Rh_2I_9$ clusters (~1.7 nm) supported on NC ($Cs_3Rh_2I_9/NC$) to form $Cs_3Rh_2I_9/NC-R$ with a little larger size (~2.2 nm) via a bottom-

up evolution route. **c** Electrochemical reduction of bulk $Cs_3Rh_2I_9$ to form $Cs_3Rh_2I_9$-R with vert large particle size via a top-down route. **d** Liquid reduction of $RhCl_3$ in aqueous to form Rh/NC with average particle size of 2.4 nm as the control catalyst. **e** HER polarization curves of $Cs_3Rh_2I_9/NC-R$, $Cs_3Rh_2I_9-R$, and Rh/NC coated on rotating glassy carbon electrode (GCE, 1600 rpm) in a chlorine-alkali electrolyte. **f** HER mass activity comparison in a chlorine-alkali electrolyte for $Cs_3Rh_2I_9/NC-R$, $Cs_3Rh_2I_9-R$, and Rh/NC electrocatalysts at an overpotential of 50 mV.

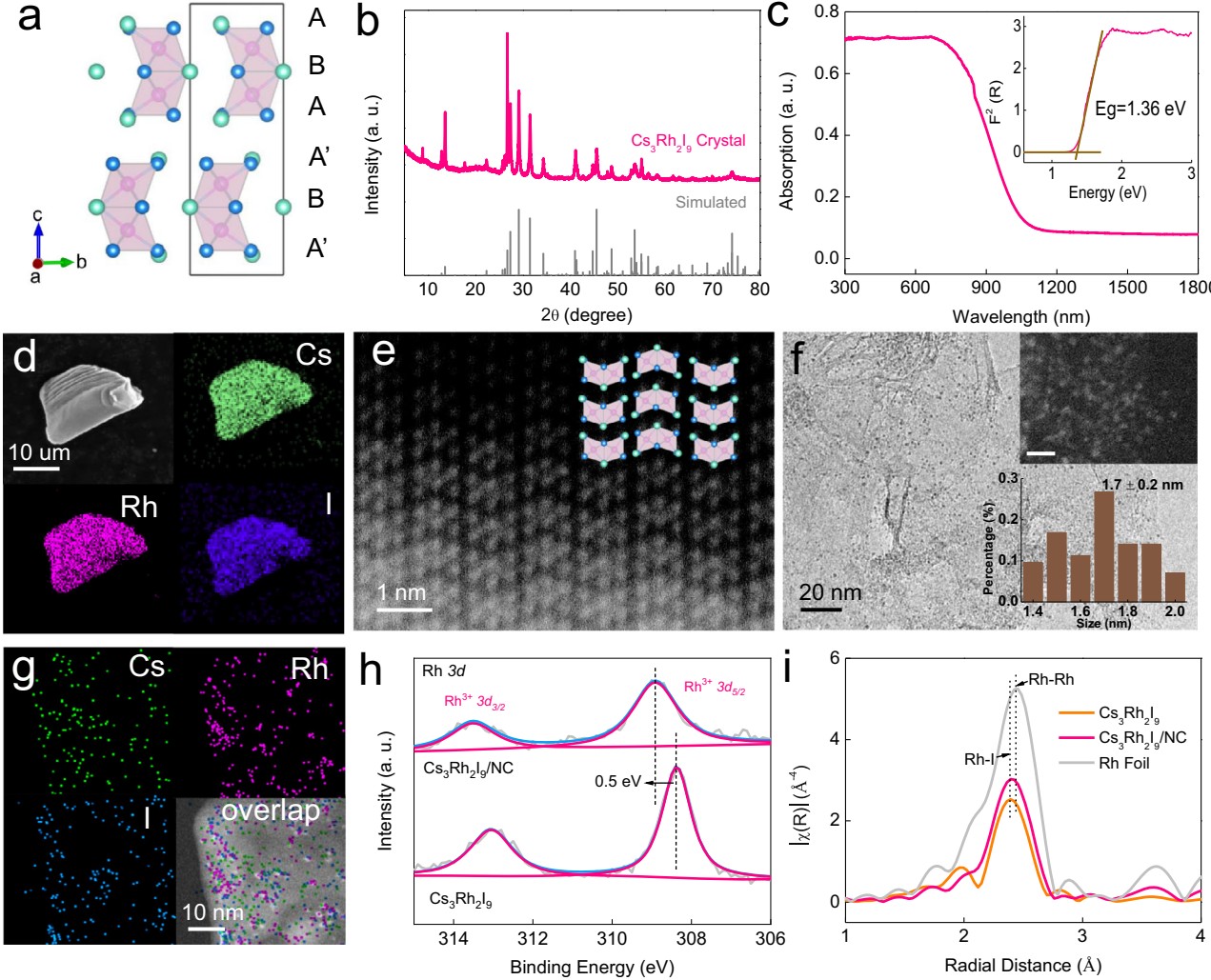

**Fig. 2 | Structure characterization of bulk Cs₃Rh₂I₉ and nanoclusters Cs₃Rh₂I₉/NC. a** Atomic structure, and (**b**) powder XRD pattern and simulated pattern of Cs₃Rh₂I₉ crystal. **c** Diffuse-reflectance UV–Vis spectrum of Cs₃Rh₂I₉. The inset shows the corresponding Tauc plot. **d** SEM-EDX mapping images of Cs₃Rh₂I₉ single crystal. **e** HAADF-STEM image of Cs₃Rh₂I₉ crystal. **f** TEM image of Cs₃Rh₂I₉ clusters supported on NC (Cs₃Rh₂I₉/NC, Rh content: 5.8 wt.%). The insets show the corresponding HAADF image (scale bar: 5 nm, up) and particle size distribution (down). **g** TEM-EDX mapping of Cs₃Rh₂I₉/NC. **h** XPS spectra of Rh 3d for Cs₃Rh₂I₉ and Cs₃Rh₂I₉/NC. **i** Fourier transform EXAFS of Rh K-edge in R-space for Cs₃Rh₂I₉, Cs₃Rh₂I₉/NC, and Rh foil.

## Results

### Synthesis and characterization of bulk and cluster Cs₃Rh₂I₉

As a new compound, Cs₃Rh₂I₉ single crystals were synthesized by the solid state reaction using CsI as the flux. Its structure was determined by single-crystal X-ray diffraction (XRD). It belongs to the hexagonal $P6_3/mmc$ space group with $a = b = 7.9648$ (14) Å, $c = 20.028$ (4) Å, $V = 1100.3$ (4) Å³, $Z = 2$, and a calculated density of $d = 5.272 \, \text{g cm}^{-3}$. Details of the atomic coordinates in the compound are shown in Supplementary Table 1, 2. Cs₃Rh₂I₉ with a zero-dimensional structure consists of alternating hexagonal CsI₃ layers in ABAA'BA' stacking sequence (Fig. 2a). Two adjacent [RhI₆]³⁻ octahedrons share three I atoms on the plane B to form a [Rh₂I₉]³⁻ dimer. The powder XRD pattern of Cs₃Rh₂I₉ is consistent with the simulated result (Fig. 2b), confirming its zero-dimensional perovskite structure[39]. Cs₃Rh₂I₉ shows a semiconductor behavior with an optical band gap of 1.36 V (Fig. 2c)[40]. Single-crystal Cs₃Rh₂I₉ is diamagnetic due to the d⁶ electronic configuration of Rh³⁺ (Supplementary Fig. 2). Energy dispersive X-ray (EDX) spectroscopy shows homogenous distribution of Cs, Rh, and I elements in crystalline Cs₃Rh₂I₉ (Fig. 2d) and EDX results from TEM show Cs, Rh and I with an atomic ratio of ~3:2:9, consistent with the stoichiometry of Cs₃Rh₂I₉ (Supplementary Fig. 3 and Table 3).

The high-angle annular dark-field scanning transmission electron microscopy (HAADF-STEM) confirms the alternating CsI₃ layer structure (Fig. 2e). Cs₃Rh₂I₉ also exhibits good stability as its structure does not change in 1.0 M HCl or KOH for 7 days (Supplementary Fig. 5).

Cs₃Rh₂I₉ can be dissolved in N, N-dimethylformamide (DMF) to form a brownish yellow solution due to the polar aprotic property of DMF (Supplementary Fig. 6). When adding water, Cs₃Rh₂I₉ precipitates without any change of structure (Supplementary Fig. 7, 8 and 10). By this means, the use of polar nitrogen-doped carbon (NC) during the precipitation process can significantly reduce the particle size to form Cs₃Rh₂I₉ nanoclusters on NC (Cs₃Rh₂I₉/NC). Figure 2f shows the uniform distribution of Cs₃Rh₂I₉ nanoclusters with a size of ~1.7 nm. No obvious particles were observed in SEM images, further confirming the uniform and ultra-small size of Cs₃Rh₂I₉ (Supplementary Fig. 9). The structure and composition of Cs₃Rh₂I₉/NC is the same as Cs₃Rh₂I₉ (Fig. 2g and Supplementary Fig. 10). The X-ray photoelectron spectroscopy (XPS) spectra show a positive core level shift of about 0.5 eV for Cs₃Rh₂I₉/NC (Fig. 2h), indicating the strong carrier effect of Cs₃Rh₂I₉ clusters on the NC[41]. The Rh−I coordination was identified by extended X-ray absorption fine structure (EXAFS; Fig. 2i). The shell at 2.40 Å represents the Rh−I scattering path for Cs₃Rh₂I₉ and

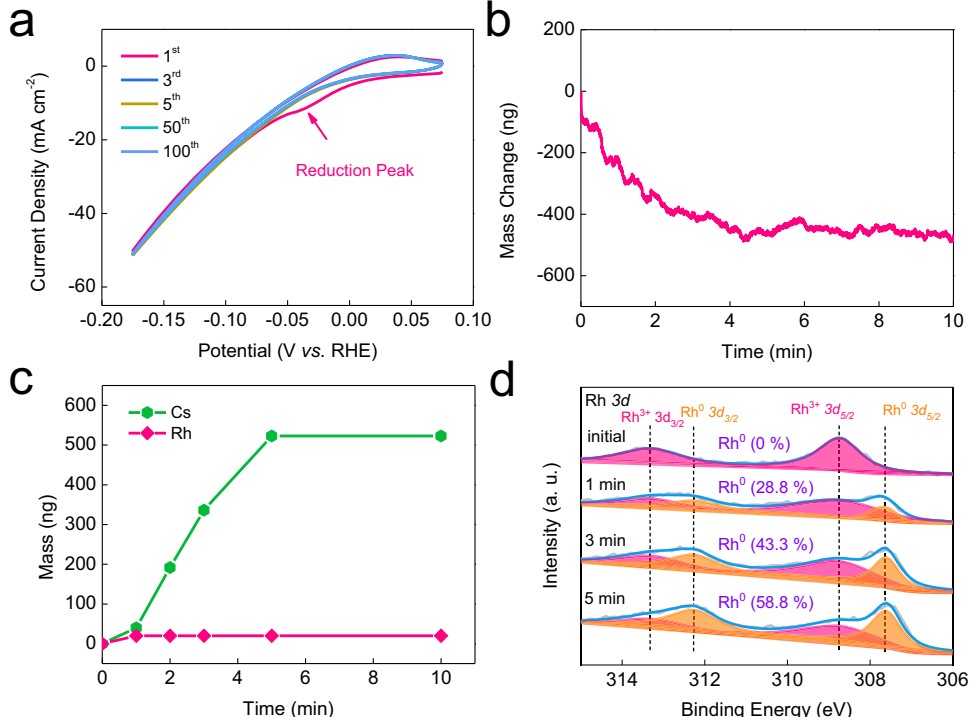

**Fig. 3 | Reconstruction of clusters Cs₃Rh₂I₉/NC into Rh nanoparticles on NC.** **a** The CV curves from 1st to 100th cycle at 100 mV s⁻¹ for Cs₃Rh₂I₉/NC in 1.0 M KOH. **b** Mass change of the Cs₃Rh₂I₉/NC electrode monitored by in situ EQCM experiment. **c** ICP-MS of the Cs and Rh contents in the electrolyte at different reduction time. **d** XPS spectra of Rh 3d for Cs₃Rh₂I₉/NC at different reduction time. The results in (**b**–**d**) were obtained at the potentiostatic measurement at −0.03 V vs. RHE.

Cs₃Rh₂I₉/NC, while the fitting results suggest that an additional Rh−Rh scattering path appears in Cs₃Rh₂I₉/NC (Supplementary Fig. 13 and Table 4). This may be caused by partial decomposition of Cs₃Rh₂I₉ nanoclusters under the high-energy measurement (Supplementary Fig. 14).

**Electrochemical reduction of Cs₃Rh₂I₉**

Under cathodic potentials, it was found that both Cs₃Rh₂I₉ clusters and single crystals are unstable and could be transformed into Rh particles. Especially, the Cs₃Rh₂I₉ nanoclusters on NC can be electrochemically reduced and assembled into Rh nanoparticles with mean particle size of 2.2 nm via a bottom-up evolution route (Fig. 1b). The reduction peak at about −0.035 V versus reversible hydrogen electrode (vs. RHE) in the first CV curve corresponds to the reduction of Rh³⁺ (Fig. 3a). The in situ Raman experiments indicate that the characteristic structure of Cs₃Rh₂I₉ disappears when the negative potential was applied (Supplementary Fig. 15). To elucidate the process of reconstruction, the reduction was conducted by the potentiostatic measurement at −0.03 V vs. RHE. The in situ electrochemical Quartz Crystal Microbalance (EQCM) experiment indicates that the quality of the electrode continuously decreases and then stabilizes after about 5 min (Fig. 3b). The Inductively Coupled Plasma Mass Spectrometry (ICP-MS) results show the content of Cs⁺ ions in the electrolyte continuously increases in the initial 5 min while Rh in the electrolyte was barely detected (Fig. 3c). And the XPS spectra shows neither Cs nor I was detected after reduction (Supplementary Fig. 16), which is also confirmed by ICP-MS results (Supplementary Table 5). Rh mass content in Cs₃Rh₂I₉/NC-R was determined by the ICP-MS to be 5.7 ± 0.8 wt.%, which is very close to the initial loading of 5.8 wt.%. This indicates all Cs₃Rh₂I₉ nanoclusters could be electrochemically reduced to metallic Rh⁰ nanoparticles. And such a reconstruction process is complete. The ex-situ XPS spectra verify the content of Rh³⁺ decreases while the content of Rh⁰ increases with the reduction time (Fig. 3d and Supplementary Fig. 17).

TEM measurements confirm the uniform distribution of Rh particles on the NC after electrochemical reduction (Cs₃Rh₂I₉/NC-R, Fig. 4a and Supplementary Fig. 18) and the Rh particle size (-2.2 nm) is evidently larger than that of Cs₃Rh₂I₉ nanoclusters (-1.7 nm), implying a bottom-up evolution route under cathodic potential. The linear electron energy loss spectroscopy analysis suggests the element in the particle is Rh (Fig. 4b). The bulk Cs₃Rh₂I₉ can also be reconstructed in this way, but it does not reach a stable state within 300 CV cycles due its large size (Supplementary Fig. 19). After reduction (Cs₃Rh₂I₉-R), its edge consists of numerous Rh particles with larger size (4.3 ± 1.2 nm) compared to the product from Cs₃Rh₂I₉/NC (Supplementary Fig. 20, 21). The Rh K-edge X-ray absorption near edge structure spectra indicates the adsorption edge of the reduced Cs₃Rh₂I₉/NC (Cs₃Rh₂I₉/NC-R) shifts to lower energy compared to the initial state (Supplementary Fig. 22). Meanwhile, the shell in EXAFS for the Cs₃Rh₂I₉/NC-R shifts to 2.45 Å (Fig. 4c), similar to that of Rh foil. The fitting results show that its Rh−Rh coordination number is only 8.0 (Fig. 4c and Supplementary Table 6), showing the small particle size. The wavelet transform (WT)-EXAFS analysis shows the first shell of Cs₃Rh₂I₉/NC-R domain at R = 2.45 Å and k = 9.90 Å⁻¹ (Fig. 4d), similar to the Rh foil (R = 2.43 Å and k = 9.80 Å⁻¹) and different from Cs₃Rh₂I₉/NC with Rh−Rh scattering path (R = 2.40 Å and k = 9.50 Å⁻¹).

Moreover, the HAADF-STEM images indicate the Rh nanoparticles in Cs₃Rh₂I₉/NC-R are rich with grain boundaries (GBs) and large lattice spacings (0.225−0.230 nm) (Fig. 4e and Supplementary Fig. 24). In sharp contrast, the particles in electrochemically reduced Cs₃Rh₂I₉ (Cs₃Rh₂I₉-R) form bulk Cs₃Rh₂I₉ show smaller plane spacings of 0.220 nm and no evident GBs could be observed (Fig. 4f and Supplementary Fig. 21). The formation of twinned Rh with large lattice spacings in Cs₃Rh₂I₉/NC-R may be ascribed to the coupling of smaller Rh clusters with high surface energy during the electrochemical reduction process. Limited by the size of the Cs₃Rh₂I₉ cluster (-1.7 nm), the formed Rh clusters could combine with neighboring clusters into

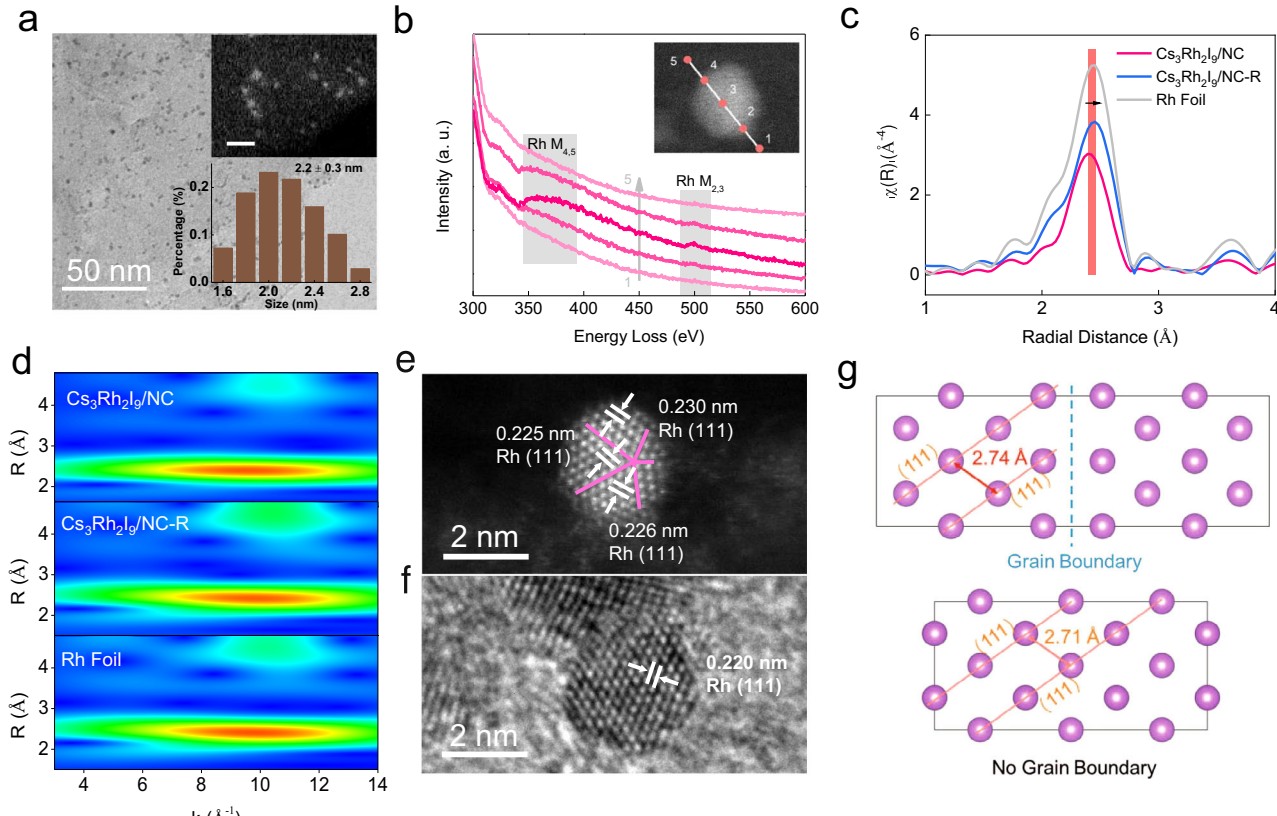

**Fig. 4 | Characterization of Cs₃Rh₂I₉/NC-R prepared by electrochemical reduction of Cs₃Rh₂I₉/NC. a** TEM image of Cs₃Rh₂I₉/NC-R, showing uniform dispersion of Rh nanoparticles. The inset shows the corresponding HAADF image (scale bar: 5 nm, up) and particle size distribution (down). **b** The linear electron energy loss spectroscopy of Cs₃Rh₂I₉/NC-R. **c** Fourier transform EXAFS of the Rh K-edge in R-space for Cs₃Rh₂I₉/NC, Cs₃Rh₂I₉/NC-R and Rh foil. **d** WT-EXAFS images of Rh for Cs₃Rh₂I₉/NC, Cs₃Rh₂I₉/NC-R and Rh foil. **e** HAADF-STEM image of Cs₃Rh₂I₉/NC-R, showing a single Rh nanoparticle with larger lattice spacing and twin-GBs. **f** HRTEM image of Rh nanoparticles (Cs₃Rh₂I₉-R) derived from the top-down electrochemical reduction of bulk Cs₃Rh₂I₉ crystal. **g** DFT models of Rh with larger lattice spacing and GBs (up panel) and Rh with regular lattice spacing (down panel).

particles (~2.2 nm) to decrease the surface energy (Supplementary Fig. 25). However, the Rh particles formed from bulk Cs₃Rh₂I₉ undergo a top-down process of bulk decomposition. They have particle sizes greater than 2 nm and thus are energetically stable enough to not combine into twin crystals. Furthermore, the twinned Rh in Cs₃Rh₂I₉/NC-R exhibits tensile stress with the enlarged lattice fringe of 0.5–5.5% (Fig. 4e, Supplementary Fig. 24). The corresponding GB was established by density functional theory (DFT) calculation (Fig. 4g) to promote tensile stress of the nearby Rh atoms (from 2.71 to 2.74 Å), suggesting the rich GBs in twinned Rh can stabilize the enlarged (111) lattice. The Rh particle with such tensile stress is considered to facilitate H₂O dissociation in alkaline HER[2].

## HER activity evaluation

The HER activity was first evaluated in 1.0 M KOH electrolyte. Before measurement, the electrode was in situ reduced under the CV between 0.075 and -0.175 vs. RHE at a scan rate of 100 mV s⁻¹ to achieve a stable stage. The Rh/NC with mean size of ~2.4 nm was synthesized by liquid chemical reduction for comparison (Supplementary Fig. 26). The Rh mass content in Rh/NC is 7 wt.%, and the HRTEM implies the lattice spacing of Rh particle is 0.220 nm, similar to that of Cs₃Rh₂I₉-R without tensile stress. As shown in Fig. 5a, the overpotential at 10 mA cm⁻² for Cs₃Rh₂I₉/NC-R composed of Rh twin nanoparticles is only 25 mV, evidently lower than those of Cs₃Rh₂I₉-R (123 mV), Pt/C (32 mV) and Rh/NC (41 mV). Moreover, the Tafel slope (30.3 mV dec⁻¹) of Cs₃Rh₂I₉/NC-R is also smaller than that of chemically reduced Rh/NC without GBs and tensile stress (34.8 mV dec⁻¹) (Supplementary Fig. 28a). It implies that the GBs and tensile stress in Rh nanoparticles have a

significant effect on boosting alkaline HER. Besides, Cs₃Rh₂I₉/NC-R exhibits high mass activity of 839.8 mA mg⁻¹_Rh, 21.6 times that of Cs₃Rh₂I₉-R and 2.4 times that of Rh/NC (Supplementary Fig. 28b). The mass loading of Cs₃Rh₂I₉/NCs was also optimized and compared (Supplementary Fig. 29). The electrochemical surface area (ECSA) is enhanced with the increased mass loading, but the optimal activity is achieved at a Rh mass loading of 5.8 wt.% (Cs₃Rh₂I₉/NC-R; Supplementary Fig. 30-32). In addition, the H₂ production faradaic efficiency for twinned Rh (Cs₃Rh₂I₉/NC-R) was confirmed to be nearly 100% through a drainage method (Supplementary Fig. 33). The HER activity of Cs₃Rh₂I₉/NC-R outperforms most of reported Rh-based electrocatalysts (Fig. 5b) and other advanced alkaline electrocatalysts (Supplementary Table 7).

The HER activity in the simulated chlorine-alkali electrolyte (3.0 M NaOH + 3.0 M NaCl) was further evaluated in a three-electrode system. As shown in Fig. 5c, the overpotentials for Cs₃Rh₂I₉/NC-R are 21, 65, and 107 mV to reach current densities of 10, 50, and 100 mA cm⁻², respectively, significantly lower than those of Cs₃Rh₂I₉-R, Pt/C, and Rh/NC. The mass activity of Cs₃Rh₂I₉/NC-R is 772.1 mA mg⁻¹_Rh at −50 mV vs. RHE, outperforming Cs₃Rh₂I₉-R (21.7 mA mg⁻¹_Rh) and Rh/NC (307.8 mA mg⁻¹_Rh) (Fig. 5d, e). The area activity normalized by ECSA of Cs₃Rh₂I₉/NC-R is 0.067 mA cm⁻², manifesting a factor of 2.0 increase than that of Rh/NC (Fig. 5e and Supplementary Fig. 35b). These results confirm the excellent intrinsic activity of Cs₃Rh₂I₉/NC-R enriched with defects. The enhanced activity may be ascribed to the reconstruction process. Cs₃Rh₂I₉/NC-R with unsaturated atom coordination and more accessible area could accelerate water adsorption and dissociation processes, thus leading to the better performance. Furthermore,

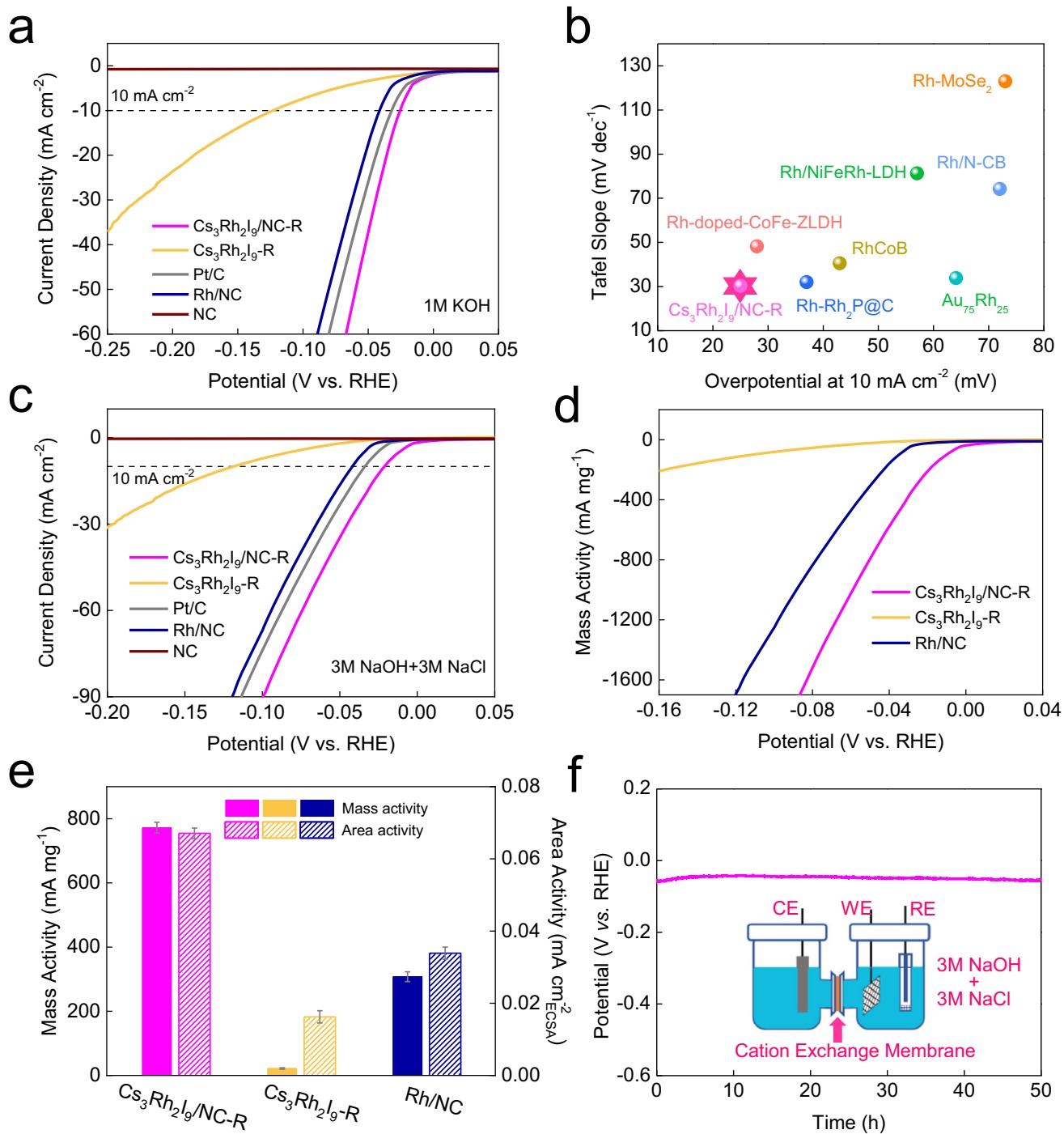

**Fig. 5 | HER activity in 1.0 M KOH and chlor-alkali electrolyte. a** LSV curves of various electrocatalysts including commercial Pt/C (Pt content: 20 wt.%) coated on rotating glassy carbon electrode (GCE, 1600 rpm) in 1.0 M KOH. The catalyst loading amount on GCE is 0.764 mg cm$^{-2}$. For Rh-based samples, the calculated Rh loading amounts on GCE are 0.045, 0.090 and 0.053 mg cm$^{-2}$ for Cs$_3$Rh$_2$I$_9$/NC-R, Cs$_3$Rh$_2$I$_9$-R, and Rh/NC, respectively. **b** Comparison of overpotential at 10 mA cm$^{-2}$ and Tafel slope for various Rh-based catalysts in 1.0 M KOH. **c** LSV curves of various electrocatalysts coated on rotating GCE (1600 rpm) in chlorine-alkali electrolyte. **d** Mass activity normalized to the mass of Rh in chlorine-alkali electrolyte. **e** Comparison of mass activity and area activity at the overpotential of 50 mV in chlorine-alkali electrolyte. All data show the mean and standard deviation through three repeated measurements. **f** Stability of Cs$_3$Rh$_2$I$_9$/NC-R at 10 mA cm$^{-2}$ in a chlorine-alkali electrolyte.

Cs$_3$Rh$_2$I$_9$/NC-R exhibits good durability with negligible activity loss after 50 h (Fig. 5f). After the durability measurement, the twin Rh nanoparticles show little agglomeration while the chemical state remains unchanged (Supplementary Fig. 37, 38).

To further disclose the origin of high activity of Cs$_3$Rh$_2$I$_9$/NC-R in the alkaline HER process, the DFT calculation was performed. Based on the above characterizations, the Cs$_3$Rh$_2$I$_9$ clusters-derived Rh nanoparticles possess larger lattice spacings and more GBs that liquid-reduced Rh nanoparticles and Rh particles from electro-reduction of bulk Cs$_3$Rh$_2$I$_9$. Thus, different Rh models with larger lattice spacings and more GBs were built to study the HER mechanism in alkaline. As shown in Supplementary Fig. 39, the increase of H−O−H angle and water molecule binding energy agrees well with the degree of distortion and crystal tension, indicating the enhanced ability of water

dissociation. Moreover, the largest H−O−H angle of 105.25° is found when water molecule is adsorbed on GBs with low coordination number, which further shows GBs would activate the H−OH bond. The water disassociation, as a pre-reaction to form an adsorbed proton, plays a more critical role in alkaline hydrogen evolution[8,42]. The linear Brønsted−Evans−Polanyi relationship between adsorption energy and dissociative kinetic barrier of $H_2O$ allows the use of binding energy of water molecule as the activity descriptor for alkaline HER. The sites at GBs and tensed Rh atoms show high $H_2O$ binding energy. The enhanced water dissociation ability in $Cs_3Rh_2I_9$/NC-R was further verified by in situ Raman spectra. Supplementary Fig. 40a shows obvious interfacial water on $Cs_3Rh_2I_9$/NC-R surface in 1 M KOH solution over the potential range from 0 to −0.3 V (vs. RHE). The peaks at 1588 and 1632 cm$^{-1}$ correspond to the G band of carbon substrate and the adsorbed water on Rh, respectively[43,44]. As the potential decreased, the intensity of H−O−H bending increases sharply. However, such a phenomenon is absent in Rh/NC synthesized by chemical reduction (Supplementary Fig. 40b). Thus, the abundant and multiply active sites on twin crystal Rh are effective for cleaving the H−OH bond, contributing to the high electrochemical performance toward HER under alkaline condition.

## Discussion

The defects-rich Rh nanoparticles with average size of ~2.2 nm were prepared by in situ electrochemical reduction of perovskite $Cs_3Rh_2I_9$ cluster via a bottom-up evolution route. The reduction was investigated by in situ EQCM, ex-situ ICP-MS, and XPS. The as-formed twin crystal Rh nanoparticles with tensile stress exhibits excellent activity and stability in alkaline hydrogen evolution reaction. In 1.0 M KOH, the $Cs_3Rh_2I_9$/NC-R catalyst showed a low overpotential of 25 mV at the current density of 10 mA cm$^{-2}$ and a small Tafel slope of 30.3 mV dec$^{-1}$. $Cs_3Rh_2I_9$/NC-R exhibits high mass activity of 839.8 mA mg$^{-1}_{Rh}$, 21.6 times that of $Cs_3Rh_2I_9$-R with bigger size and 2.4 times that of liquid-reduced Rh/NC. In a chlor-alkali electrolyte, the area activity of $Cs_3Rh_2I_9$/NC-R (0.067 mA cm$^{-2}_{ECSA}$ at −50 mV vs. RHE) manifests a factor of 4.1 and 2.0 activity increase compared to $Cs_3Rh_2I_9$-R and Rh/NC, respectively. Moreover, it exhibits good durability with negligible activity loss for 50 h HER measurement. The DFT calculation revealed that Rh nanoparticles of $Cs_3Rh_2I_9$/NC-R enriched with multiply catalytic sites could accelerate the activation of adsorbed water molecule, thereby smoothing the whole alkaline HER. The study presents new insights into preparing small-sized nanoparticles via in situ electrochemical reconstruction for energy electrocatalysis.

## Methods

### Synthesis of $Cs_3Rh_2I_9$ crystal

The $Cs_3Rh_2I_9$ crystal was prepared via solid state reaction. 50 mg rhodium, 185 mg iodine and 1.5 g CsI were mixed by grinding in Ar glovebox to prevent the influence of water. CsI was served as both raw material and flux. The above powder was annealed at 800 °C for 2000 min in the evacuated quartz tube and cooled to room temperature at the rate of 3 °C min$^{-1}$. Finally, the product was washed with deionized water for several times and dried in vacuum at room temperature.

### Synthesis of NC

NC was prepared by the sol−gel method according to our previous report.[41] It started from mixing 1.8 mL formaldehyde and 1 g melamine in 20 mL deionized water under stirring at 50 °C for 1 h. Then, 4.5 g $MnCl_2$ and 6 g PEG were added into the above solution and continuously stirred at room temperature to form a uniform sol. The sol was transferred into a culture dish and dried at 80 °C for 24 h to form a gel precursor. The gel was cut into small slices and pre-carbonized at 400 °C for 2 h in an Ar atmosphere. After grounding the precursor into powder, it was annealed at 900 °C with a heat rate of 3 °C min$^{-1}$ in the

Ar atmosphere. The final product NC was obtained by acid treatment to remove the metal impurities.

### Synthesis of nanoclusters $Cs_3Rh_2I_9$ supported on NC ($Cs_3Rh_2I_9$/NC)

10 mg $Cs_3Rh_2I_9$ crystal was added into 40 ml N, N-dimethylformamide (DMF) and stirred at 60 °C to form the brown solution. Then, 10 mg NC was added and the mixture was continually stirred for 1 h. Subsequently, the above mixture was slowly added into 200 ml $H_2O$ under fiercely stirring. The induced polar protic solvent with slow $S_N2$ reaction kinetics results in the precipitation of $Cs_3Rh_2I_9$[45]. The $Cs_3Rh_2I_9$/NC was collected by suction filtration and washed by water and ethanol for several times. And it was dried in vacuum at 60 °C for 12 h. $Cs_3Rh_2I_9$/NC with different Rh contents (x wt.%, x = 2.7, 4.3, 5.8, or 7.1) was prepared by changing mass loading of $Cs_3Rh_2I_9$, and x corresponds to the Rh content. The optimized $Cs_3Rh_2I_9$/NC (5.8 wt.%) is named as $Cs_3Rh_2I_9$/NC in the main text and SI if no specific note was mentioned. The re-precipitated $Cs_3Rh_2I_9$ without adding NC was obtained as the control sample.

### Synthesis of Rh/NC by liquid reduction

$RhCl_3$ (7.1 mg) was dissolved in 20 mL N-methyl pyrrolidone (NMP), followed by the addition of 46.5 mg NC. After stirring for 1 h, 0.05 g sodium borohydride in NMP was added into the above solution drop by drop. After stirring for 20 h, Rh/NC was collected by suction filtration and then annealed in 10% $H_2$/Ar at 300 °C for 30 min. The elemental analysis shows the mass content of Rh in Rh/NC is 7.0 wt.%.

### Characterization

Single-crystal XRD data was obtained on a Bruker D8 QUEST diffractometer with Mo-$K_\alpha$ (λ = 0.71073 Å) radiation at 300 K. The crystal structure was solved and refined using APEX3 program. Powder XRD was performed on a Bruker D8 Advance diffractometer equipped with mirror-monochromatized source of Cu Kα radiation (λ = 0.15406 nm). The ultraviolet−visible (UV−Vis) light diffuse-reflectance spectra were measured on a UV-4100 spectrophotometer operating from 2000 to 300 nm at room temperature and the $BaSO_4$ powder was used as a 100% reflectance standard. Low-temperature electrical resistivity was measured using a Physical Properties Measurements System (PPMS-Dyna Cool, Quantum Design). SEM was conducted on the JSM-7800F. TEM was conducted on the JEM-2100F and Talos F200X G2. HAADF-STEM and EELS measurements were obtained from aberration-corrected TEM (Hitachi HF5000) and JEOL Triple-C TEM. The chemical states were investigated by X-ray photoelectron spectroscopy (XPS, Thermo Fisher Scientific ESCA Lab 250Xi spectrometer) with focused monochromatic Al Kα radiation (1486.6 eV, 150 W; 500 μm diameter of irradiated area). Ion concentration in electrolyte was determined by the inductively coupled plasma mass spectrometry (XII, Thermo Fisher Scientific). X-ray absorption fine-structure spectroscopy (XAFS) was performed at the Materials Research Collaborative Access Team (MRCAT), Sector 10-BM line at the Advanced Photon Source at Argonne National Laboratory[46]. Data were processed and fitted using Athena and Artemis for the IFEFFIT suite[47,48]. All spectra were prepared for Fourier Transform using a Hanning window ranging from 2.0 Å$^{-2}$ < k < 14 Å$^{-1}$ with dk = 2 Å$^{-1}$ and simultaneously fitted in k, $k^2$, and $k^3$ weightings using a Hanning window of 1.8 Å < R < 2.8 Å with dR = 0.2 Å. Wavelet transforms were obtained from processed data using Larch[49].

### Hydrogen evolution reaction experiments

The electrochemical performance was conducted on the three-electrode system using graphite rod as counter electrode and Hg/HgO as the reference electrode at 25 °C. 5 mg catalyst and 25 uL Nafion (5 wt.%) were added into 475 uL ethanol to form the homogeneous catalyst ink. And the working electrode was prepared by adding 15 uL

ink on a rotating GCE (area: 0.1963 cm$^{-2}$). The catalyst loading amount on GCE is 0.764 mg cm$^{-2}$. And the calculated Rh loading amounts on GCE are 0.045, 0.090, and 0.053 mg cm$^{-2}$ for Cs$_3$Rh$_2$I$_9$/NC-R, Cs$_3$Rh$_2$I$_9$-R, and Rh/NC, respectively. Moreover, the commercial Pt/C (Pt content: 20 wt.%) was also used as the control electrocatalyst. The HER measurements were performed in the Ar-saturated electrolyte at the rotating speed of 1600 rpm. Before measurement, the electrode was activated (reduced) under the CV between 0.075 and -0.175 vs. RHE at a scan rate of 100 mV s$^{-1}$ for 100 cycles. The linear sweep voltammetry (LSV) curves were recorded at the scan rate of 5 mV s$^{-1}$ with the IR-compensation 90%. The ECSA was obtained from the equation ECSA = $C_{dl}/C_s$, where the electrochemical double layer capacitance ($C_{dl}$) was obtained from the CV measurement at different scan rates and the specific capacitance ($C_s$) was 0.4 μF cm$^{-2}$ in 1.0 M KOH. In chlorine-alkali electrolyte (3.0 M NaOH + 3.0 M NaCl), the Hg/HgO electrode was protected by the salt bridge (1.0 M KOH).

The durable measurement was conducted in a two-compartment cell using the cation exchange membrane as the separator. And the working electrode was prepared by dropping the ink on the carbon cloth (catalyst loading amount: 1 mg cm$^{-2}$).

### Reversible hydrogen electrode (RHE) calibration

The calibration was performed in the H$_2$-saturated electrolyte using Pt foils as counter electrode and working electrode. CV measurements were carried out at the scan rate of 1 mV s$^{-1}$. The average potentials at the current of zero were set as the thermodynamic potential of RHE.

### Electrochemical quartz crystal microbalance experiment

The EQCM experiment was conducted on the QSense Explorer instrument (Biolin Scientific AB, Sweden). The working electrode was prepared by spin-coating the catalyst ink on the Au-coated quartz crystal disk (5 MHz). In situ EQCM was measured simultaneously with potentiostatic measurement at −0.03 V vs. RHE.

### Hydrogen production faradic efficiency

The faradic efficiency was determined by the drainage method at 25 °C. A constant current was applied on the electrode, and the electrocatalysis time and the volume of evolved hydrogen were recorded synchronously. Each experiment was repeated three times. The experimental content of produced hydrogen was calculated by the Ideal Gas Law and the theoretical content was determined by the Faraday Law.

### Computational methods

All theoretical calculations were performed using DFT, as is implemented in the Vienna ab initio simulation package (VASP). The electron exchange and correlation energy functionals are treated using the generalized gradient approximation, as is captured using the Perdew-Burke-Ernzerh functional (GGA-PBE). Iterative solutions of the Kohn–Sham equations were done using a plane-wave basis set defined using a kinetic energy cutoff of 500 eV. The k-point sampling was obtained from the Monkhorst–Pack scheme with a (6 × 6 × 1) mesh. respectively. Rh (001) surface structures with lattice tensile of 1.5% and 3% were built and optimized to simulate active sites with different crystal strain. Surface structures with Rh (110) GB and standard Rh (001) were chosen to reflect the properties of GB and monocrystal Rh, respectively. Water bonding energy ($\triangle E_{H_2O}$) was calculated by the following equation:

$$\triangle E_{H_2O} = E_{H_2O} + E_{sub} - E_{total}$$

whereas $E_{total}$ refers to energy of substrate with corresponding adsorbate, $\triangle E_{H_2O}$ refer to energy of one H$_2$O molecule, $T$ is selected by room temperature (298 K), and ZPE refers to zero point energy.

## Data availability

All data generated or analyzed during this study are included in this published article and its supplementary information file. Source data are provided with this paper.

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

## Acknowledgements

The authors are grateful to the financial support from the National Natural Science Foundation of China (92163117 for J.W., 52072389 for J.W.), Science and Technology Commission of Shanghai Municipality (22DZ1205600 for J.W., 20520760900 for J.W., 21ZR1473300 for W.Z.), State Key Laboratory of ASIC & System (2020KF002 for J.W.) and Shanghai Science and Technology Innovation Action Plan (20DZ1204400 for F.H.). Analytical Instrumentation Center (No. SPSTAIC10112914), SPST, ShanghaiTech University. J.W. thanks the Program of Shanghai Academic Research Leader (20XD1424300) for financial support. MRCAT opera-tions are supported by the Department of Energy and the MRCAT mem-ber institutions. This research used resources from the Advanced Photon Source, a U.S. Department of Energy (DOE) Office of Science User Facility operated for the DOE Office of Science by Argonne National Laboratory under Contract No. DE-AC02-06CH11357. Q.L. and K.S. thank the JST-CREST and ER-C MORE-TEM synergy projects.

## Author contributions

G.L., Z.Z., and Q.J. contributed equally to this work. F.H. and J.W. supervised and led the project. G.L., Z.Z., and Y.M. performed materials synthesis. Q.J. conducted theoretical calculations. G.L. and T.W. char-acterized the materials. C.S. and W.C. helped with XAFS. H.P., H.Z., and Z.L. helped with XPS measurement. Q.L. and K.S. collected the atomic-scale STEM images of Rh nanoparticles. Y.Z. conducted EQCM experi-ment. G.L. and S.K. carried out electrochemical experiments. G.L. wrote the manuscript. W.Z., F.H., and J.W. revised the manuscript.

## Competing interests

The authors declare no competing interests.
