## [Peer review file · Nature Communications]

REVIEWER COMMENTS

Reviewer #1 (Remarks to the Author):

This manuscript deals with the preparation of Rh nanoparticles containing grain boundaries (GBs) via the electrochemical reduction of perovskite Cs₃Rh₂I₉. A major claim by the authors is that this reduction process has an advantage of obtaining defective Rh nanoparticles with GBs and their HER activity can be significantly enhanced by the presence of GBs. Although the notable improvement of catalytic activity by an order of magnitude in HER current density is highly appreciated, the following issues prevent us from recommending publication of this manuscript.

1. Even though the provision of clear-cut evidence proving the presence of GBs in Rh nanoparticles is of great significance in this study, no reliable images for GBs are provided in the manuscript. In particular, the authors repeatedly show exactly the same HAADF image in both Figure 1b and Figure 4e. Unless a sufficient number of clear images for GBs are given to confirm the existence of GBs in the nanoparticles, very few readers would agree with the authors' claim (for example, see the previous report, ACS Catal. 10, 12575-1258 (2020)).

2. Furthermore, the GBs in Figure 4e do not really appear to be grain boundaries. They are rather very likely to be stacking faults. In FCC/HCP metals, such stacking faults frequently form during nanoparticle synthesis at low temperature (see ref. 36 in this manuscript). It is thus necessary to clarify whether the Rh nanoparticles contain GBs or not.

3. Although the presence of GBs has been already reported to be beneficial in CO₂ reduction electrocatalysis, no direct correlation between the GBs and the enhancement of the HER activity is shown by the DFT calculations in this study. Readers would wonder why the tensile strain induce by GBs in Rh can significantly improve the HER activity. Moreover, the calculation on the binding energy of water molecule in Supplementary Fig. 37 does not appear to be logically reasonable to explain the role of GBs. From this viewpoint of the authors, if we could obtain any catalyst with the highest binding energy, it would show the best HER activity, which seriously contradicts the well-known 'Sabatire principle'. Even though the effect of tensile strain has been discussed in Ag (ref. 36 in this manuscript), more rigorous and systematic calculations should be carried out in Rh for readers.

4. It should be noted that nitrogen-doped carbon (NC) was used as a support material in this study. Consequently, there is concern about the formation of single-atom Rh during the electrochemical reduction of perovskite Cs₃Rh₂I₉. Although single-atom Rh is not directly captured in the HAADF image Figure 4a, it is worthwhile to examine its presence on the NC support for clarification.

Reviewer #2 (Remarks to the Author):

Bottom-up Evolution of Perovskite Clusters into High-activity Rhodium Nanoparticles Enriched with Grain Boundary

The paper reports the synthesis, characterisation, and comprehensive electrochemical characterisation of free-standing and immobilised on carbon-related support Rh-nanoparticles. The authors proposed in situ electrochemical reduction of Cs_3PdI_9 precursor as a general method which seems to help them achieve a good distribution of Rh nanoparticles on the support. This is a good proposal despite solid-state synthesis $\text{Cs}_3\text{Rh}_2\text{I}_9$ is highly complicated and few labs (and even less industrial users) are equipped with ampoule sealing apparatus. Still from understanding the role of surface passivation by surfactant the method has its merits (keeping in mind that the cost of CsI / I is negligible compared with the cost of Rh). The authors benchmarked the NP prepared by the proposed approach with Pt/C but seem to neglect the materials from a traditional method for the synthesis of Rh-nanoparticles via RhCl_3 route. Overall, this is a very consistent work and I like the thorough approach authors took to support their results.

However, the authors try to make the case that grain boundaries play the key role in improving the catalytic performance of nanoparticles (when compared with unsupported nanoparticles). This is something I was not convinced completely. I am not sure why authors decided to go a rather treacherous route and insist that grain boundaries are the reason for improved performance given that the work is quite consistent and would merit publication without this speculative (and hard to prove) idea. Well, since the authors are convinced that grain boundaries are essential, I would like them to provide more convincing evidence and thus, there are some additional experiments I would like them to conduct: the materials should be benchmarked against Rh-NP prepared by reduction of RhCl_3 with sodium borohydride.

Still, despite the contested issue of the catalytic properties originating from grain boundaries I recommend publication of this article after technical issues are addressed:

1. The authors try to give the impression to the reader that Pt/C is the catalyst of choice for the HER in alkaline electrolyzers. This is not true, it is a key catalyst in PEM electrolyzers, i.e. acidic electrolytes. In alkaline conditions, Ni-Mo or Ni-Fe-Mo alloys even Ni (foam) are widely used in commercial electrolyzers. This is because the limiting stage is OER and the activity of HER does not matter that much on overall performance of the stack. I would like to the authors to soften their statement a little bit and mention some good work on alkaline electrolyzers. The notion that Pt/C is used in alkaline electrolysis is not helpful as it propagates the erroneous perception and confuses unexperienced readers. This is something that should be avoided given general readership of Nature Comms.
2. I do not understand why $\text{Cs}_3\text{Rh}_2\text{I}_9$ got dispersed in rather polar DMF but can be quenched (precipitated) from the DMF with water. Authors, please provide explanation. More details and

discussion should be provided here as it would allow for a better reproducibility and understanding of the experiments.

3. Please clarify whether Rh 3d spectrum (bottom) recorded on the initial Cs₃Rh₂I₉ single crystals or non-immobilized Cs₃Rh₂I₉ (i.e. NC-free particles – called by authors “monocrystalline Rh”). If it is recorded on single crystals than the XPS of unsupported Cs₃Rh₂I₉ should be provided as it seems that support makes a significant difference and questions the assumption that grain boundaries play the key role in catalytic performance .

4. I found the potential value of -0.035 V for the reduction wave in the initial CV scan too low given that Rh³⁺ → Rh standard reduction potential is in the order 0.6 V. To me it more consistent with surface oxidation. Same seems to apply to the NC-free Cs₃Rh₂I₉ as well as shown in the SI. Is there explanation for that?

5. Please provide Raman spectrum recorded on pure Cs₃Rh₂I₉ which led “monocrystalline Rh”). The in operando spectra are very noisy and there is no peak assignment. I found it very, very strange that the reduction is almost instant especially at such low potential.

6. From the Fig. S20 it seems that unsupported Rh-particles are quite large, furthermore, plenty of grain boundaries are visible which seems to defy the hypothesis that improved characteristic of supported materials are due to grain boundaries. The particles are highly aggregated as well which given that button electrode is used makes it impossible to conclude whether it is the role of particles size, preferred orientation, support, or grain boundaries. This echoes the results presented in Fig. 4e and 4f which is based on two HRTEM images. Why are the authors so convinced that GB play dominant role? While monocrystalline NP are clearly 111 (Fig. 4f), I am not convinced from Fig4e and Fig.S25 that the Rh(111) is formed for supported NPs. Were there any SAED recorded to prove it? The bottom line is that the authors used a local method such as HRTEM (which may not be representative of the entire sample) to justify their hypothesis that grain boundaries play key role.

7. I wonder if authors prepared Rh-NP from RhCl₃ (mentioned in the methods) why did not they used them as a benchmark? As mentioned above Pt/C is not a useful benchmark here as it is not used in commercial alkaline electrolyzers and it would be more useful to compare between Rh-NP prepared by traditional method (RhCl₃) and the one proposed by the authors (Cs₃Rh₂I₉). This is more important as the article is interesting from fundamental science of view (I doubt solid state method for synthesis of Cs₃Rh₂I₉ is practical to be adopted). Therefore, It would be good to benchmark against RuCl₃-made NPs whether immobilised on NC or not. I understand that the authors showed in Fig. 5b the results from other groups but such results may be influenced by numerous factors such as the nature of support which seems to play substantial role.

Just in case that it is lost in the text above authors please add the results of electrochemical and TEM assessment of NP prepared by reduction of RhCl₃ with sodium borohydride.

Overall, I disagree with authors that “The enhanced activity may be ascribed to the polycrystalline Rh with tensile stress” the evidence presented in their work seem to point to the role of the substrate (which improved dispersibility and high active site exposure) and/or size of nanoparticles. Still, I believe this is a solid and well-presented manuscript and the article is suitable for Nature Comms. When the technical points are addressed.

Reviewer #3 (Remarks to the Author):

The authors reported the Rh nanoparticles with abundant GBs, and detailly elucidated the synthetic process and analyzed the formed mechanism of GB. In particular, the top-down and bottom-up methods for Rh nanoparticles preparation were compared by many physical characterization techniques. As expected, the GB-enriched Rh nanoparticles demonstrated the excellent catalytic activity, which is superior to monocrystalline Rh without GBs. It claims that such activity results from the boosted water dissociation kinetics by the GB with strain stress. However, there are many critical issues (see following) that prevent me from recommending this paper to publish in the high-quality journal.

1. The authors claim that the GB-enriched Rh nanoparticles were prepared by the bottom-up methods based on the NC used as support. Although many reported literatures have similar description, namely, there is a substrate as a nucleation site that is considered as bottom-up methods. However, the specific process of electrochemical reduction of Cs₃Rh₂I₉ perovskite clusters is no basically comprehensive research.
2. There is no experimental investigation to verify the enhanced water dissociation kinetics. The DFT calculations is only the theory.
3. According to the XPS and EDX with limited deflection ability, it is not certain that there is no Cs and I elements in Rh nanoparticles. Thus, the HER activity origin needs to be further investigated because the synergistic effect between elements is very important in enhancing HER performance.
4. In term of DFT calculations, especial the constructed atom model, authors did not consider the role of NC in HER process, but both XPS and XAS results evidenced the strong electronic interactions between Rh and NC.
5. In experimental method, there is a mistake. The RuCl₃ should be substituted with RhCl₃ in “synthesis of Rh/NC”.

Reviewer #1 (Remarks to the Author):

This manuscript deals with the preparation of Rh nanoparticles containing grain boundaries (GBs) via the electrochemical reduction of perovskite $\text{Cs}_3\text{Rh}_2\text{I}_9$. A major claim by the authors is that this reduction process has an advantage of obtaining defective Rh nanoparticles with GBs and their HER activity can be significantly enhanced by the presence of GBs. Although the notable improvement of catalytic activity by an order of magnitude in HER current density is highly appreciated, the following issues prevent us from recommending publication of this manuscript.

1. Even though the provision of clear-cut evidence proving the presence of GBs in Rh nanoparticles is of great significance in this study, no reliable images for GBs are provided in the manuscript. In particular, the authors repeatedly show exactly the same HAADF image in both Figure 1b and Figure 4e. Unless a sufficient number of clear images for GBs are given to confirm the existence of GBs in the nanoparticles, very few readers would agree with the authors' claim (for example, see the previous report, ACS Catal. 10, 12575-12581 (2020)).

Response: We appreciate your comments very much. We have been added enough HAADF-STEM images for $\text{Cs}_3\text{Rh}_2\text{I}_9/\text{NC-R}$ enriched with GBs (Figure R1) as Supplementary Fig. 25 in the Supplementary Information. All these images provide strong evidence that the GBs are present in lots of Rh nanoparticles for the $\text{Cs}_3\text{Rh}_2\text{I}_9/\text{NC-R}$ sample. Figure 4e in the main text was replaced by Figure R1a.

Figure R1 HAADF-STEM images of $\text{Cs}_3\text{Rh}_2\text{I}_9/\text{NC-R}$.

2. Furthermore, the GBs in Figure 4e do not really appear to be grain boundaries. They are rather very likely to be stacking faults. In FCC/HCP metals, such stacking faults frequently form during nanoparticle synthesis at low temperature (see ref. 36 in this manuscript). It is thus necessary to clarify whether the Rh nanoparticles contain GBs or not.

Response:

Thank you very much for your careful check of our manuscript. In this work, the Rh particles are formed by in-situ electrochemical reduction of $\text{Cs}_3\text{Rh}_2\text{I}_9$ perovskite clusters. The average size of initial $\text{Cs}_3\text{Rh}_2\text{I}_9$ clusters is only 1.7 nm. However, the average size of Rh particles increased to 2.2 nm after reduction due to the coupling of in-situ-formed ultra-small Rh clusters with high surface energy, which could lead to the formation of twin grain boundaries. Besides, lots of STEM images show the obvious angles of lattice planes in the twined particles, suggesting the Rh nanoparticles contain grain boundaries.

3. Although the presence of GBs has been already reported to be beneficial in CO₂ reduction electrocatalysis, no direct correlation between the GBs and the enhancement of the HER activity is shown by the DFT calculations in this study. Readers would wonder why the tensile strain induce by GBs in Rh can significantly improve the HER activity. Moreover, the calculation on the binding energy of water molecule in Supplementary Fig. 37 does not appear to be logically reasonable to explain the role of GBs. From this viewpoint of the authors, if we could obtain any catalyst with the highest binding energy, it would show the best HER activity, which seriously contradicts the well-known ‘Sabatire principle’. Even though the effect of tensile strain has been discussed in Ag (ref. 36 in this manuscript), more rigorous and systematic calculations should be carried out in Rh for readers.

Response:

We sincerely agree with you that a common understanding of adsorption strength follows the “Sabatire principle”. However, in HER, water is used as reactant, instead of intermediate. Therefore, the selection of “moderate adsorption” of water as the most appropriate adsorption strength toward alkaline HER is not appropriate. Besides, as reported in previous study, the adsorption energy and dissociative kinetic barrier of H₂O have a linear Brønsted-Evans-Polanyi (BEP) relationship (Nat. Chem. 2009, 1, 37; Nat. Commun. 2022, 13, 1270). Thus, the adsorption energy of H₂O can be used as an activity descriptor for the kinetic barrier of water dissociation. The higher H₂O binding energy is, the lower the barrier for H₂O dissociation will be. In this work, the binding energy of water molecule on Rh is 0.27 eV, and the value increases to 0.33 eV when water adsorbs on Rh with 3% tension (Supplementary Fig. 38), indicating the enhanced water dissociation ability of Rh with strain. In the structural models of adsorbed water molecule on Rh, the largest H–O–H angle of 105.25° is found when water molecule is adsorbed on GBs, which further shows GBs acting as sites would activate the H–OH bond. To make it easier for readers to understand, more discussion has been added in the main text:

The linear Brønsted-Evans-Polanyi relationship between adsorption energy and dissociative kinetic barrier of H₂O allows the use of binding energy of water molecule as the activity descriptor for alkaline HER. The sites at GBs and tensed Rh atoms show high H₂O binding energy. The enhanced water dissociation ability in Cs₃Rh₂I₉/NC-R was further verified by in situ Raman spectra. Supplementary Fig. 39a shows obvious interfacial water on Cs₃Rh₂I₉/NC-R surface in 1M KOH solution over the potential range from 0 to -0.3 V (vs. RHE). The peaks at 1588 and 1632 cm⁻¹ correspond to the G band of carbon substrate and the adsorbed water on Rh, respectively^{45, 46}. As the potential decreased, the intensity of H-O-H bending increases sharply. However, such phenomenon is absent in Rh/NC without GBs (Supplementary Fig. 39b). Thus, the abundant and multiply active sites on twin crystal Rh can be effective for cleaving the H-OH bond, contributing to the high electrochemical performance toward HER under alkaline condition.

4. It should be noted that nitrogen-doped carbon (NC) was used as a support material in this study. Consequently, there is concern about the formation of single-atom Rh during the electrochemical reduction of perovskite Cs₃Rh₂I₉. Although single-atom Rh is not directly captured in the HAADF image Figure 4a, it is worthwhile to examine its presence on the NC support for clarification.

Response:

Thank you very much for the valuable comments. The extended X-ray absorption fine structure (EXAFS) is sensitive to chemical and coordination states of element and can provide coordination number and average bond length between single atom and nearby atoms. As shown in Supplementary Fig. 24, Fourier transform EXAFS of Rh K-edge in R-space for Cs₃Rh₂I₉/NC-R exhibits the shell at 2.45 Å, similar to that of Rh foil. Fitting result proves the shell corresponds to the Rh-Rh scattering path with coordination number of 8.0. And no Rh-N(C) scattering path was detected. Besides, single-atom Rh is not directly captured in the HAADF image. Therefore, only Rh particles are formed in Cs₃Rh₂I₉/NC-R.

Reviewer #2 (Remarks to the Author):

The paper reports the synthesis, characterisation, and comprehensive electrochemical characterisation of free-standing and immobilised on carbon-related support Rh-nanoparticles. The authors proposed in situ electrochemical reduction of Cs₃Rh₂I₉ precursor as a general method which seems to help them achieve a good distribution of Rh nanoparticles on the support. This is a good proposal despite solid-state synthesis Cs₃Rh₂I₉ is highly complicated and few labs (and even less industrial users) are equipped with ampoule sealing apparatus. Still from understanding the role of surface passivation by surfactant the method has its merits (keeping in mind that the cost of CsI / I is negligible compared with the cost of Rh). The authors benchmarked the NP prepared by the proposed approach with Pt/C but seem to neglect the materials from a traditional method for the synthesis of Rh-nanoparticles via RhCl₃ route. Overall, this is a very consistent work and I like the thorough approach authors took to support their results.

Response:

Thanks very much for your positive comments and support on our work. We have revised the manuscript according to your valuable comments. Detailed discussions are listed in the following point-by-point response. Hopefully the revised edition could satisfy you.

1. However, the authors try to make the case that grain boundaries play the key role in improving the catalytic performance of nanoparticles (when compared with unsupported nanoparticles). This is something I was not convinced completely. I am not sure why authors decided to go a rather treacherous route and insist that grain boundaries are the reason for improved performance given that the work is quite consistent and would merit publication without this speculative (and hard to prove) idea. Well, since the authors are convinced that grain boundaries are essential, I would like them to provide more convincing evidence and thus, there are some additional experiments I would like them to conduct: the materials should be benchmarked against

Rh-NP prepared by reduction of RhCl₃ with sodium borohydride.

Still, despite the contested issue of the catalytic properties originating from grain boundaries I recommend publication of this article after technical issues are addressed:

Response:

Thanks for Reviewer's comprehensive suggestion to this work. In this work, we developed a bottom-up evolution route to prepare twinned Rh nanoparticles with high-density and exposed grain boundaries (GBs) *via* in-situ electrochemical reduction of Cs₃Rh₂I₉ perovskite clusters. However, in the initial stage of this work, we only tried to evaluate the HER activity of Cs₃Rh₂I₉/NC. The reduction peak in the first CV curve and increased particle size of loaded nanoparticle after reduction aroused our curiosity. Therefore, in-situ electrochemical Quartz Crystal Microbalance experiment, Inductively Coupled Plasma Mass Spectrometry and X-ray photoelectron spectroscopy were taken to elucidate the process of the electrochemical reduction. And high angle annular dark-field scanning transmission electron microscopy showed obvious GBs in reduced Cs₃Rh₂I₉/NC (Cs₃Rh₂I₉/NC-R).

Nanocrystals with twinned structure are considered as advanced catalysts due to stressed grain and boundaries from surface mismatches. The existing structural perturbation could lead to the optimization of the electronic state, the adsorption characteristics and the reaction kinetics of the catalyst. For examples, a GB-rich metallic copper showed much higher electrochemical carbon dioxide reduction activity than the copper without GB due to the enhanced CO binding and C–C coupling properties (J. Am. Chem. Soc. 2020, 142, 6878); the catalytic rate was positively correlated with GB density in Pd catalyst (Science 2021, 373, 1518) and the GBs in monolayer molybdenum disulphide acted as new highly active and tunable catalytic sites for HER (Nat. Commun. 2019, 10, 1348). Therefore, when we found the high-density and exposed GBs in Cs₃Rh₂I₉/NC and its high alkaline HER performance, we tried to prove that the GB are responsible for the enhanced activity. In 1.0 M KOH, the GB-rich Cs₃Rh₂I₉/NC-R exhibited high mass activity of 846 mA mg⁻¹_{Rh} at the overpotential of 50 mV, which is about 22.3 times that of GB-free Cs₃Rh₂I₉-R. To

compare their intrinsic activity, the current was also normalized to the electrochemical active surface area (ECSA). Cs₃Rh₂I₉/NC-R showed large current density of 0.065 mA cm⁻²_{ECSA} at the overpotential of 50 mV, much high than the 0.028 mA cm⁻²_{ECSA} of Cs₃Rh₂I₉-R. Besides, DFT calculation confirmed GBs can act as sites to activate the H–OH bond, which plays a critical role in boosting alkaline hydrogen evolution.

As reviewer pointed, Rh/NC prepared by chemical reduction of RhCl₃ with sodium borohydride should serve as a benchmark. The electrochemical performance of Rh/NC in both 1.0 M KOH and chlorine-alkali electrolyte have been added in the main text. Combined theoretical calculations with previous report, GBs are proved to play the key role in enhanced the electrolysis.

2. The authors try to give the impression to the reader that Pt/C is the catalyst of choice for the HER in alkaline electrolyzers. This is not true, it is a key catalyst in PEM electrolyzers, i.e. acidic electrolytes. In alkaline conditions, Ni-Mo or Ni-Fe-Mo alloys even Ni (foam) are widely used in commercial electrolyzers. This is because the limiting stage is OER and the activity of HER does not matter that much on overall performance of the stack. I would like to the authors to soften their statement a little bit and mention some good work on alkaline electrolyzers. The notion that Pt/C is used in alkaline electrolysis is not helpful as it propagates the erroneous perception and confuses unexperienced readers. This is something that should be avoided given general readership of Nature Comms.

Response:

Thanks very much for your valuable suggestion. To avoid the misunderstandings for readers, we have softened our statement about Pt/C, and focused more on the comparison of Cs₃Rh₂I₉/NC-R to Cs₃Rh₂I₉-R and Rh/NC. Besides, the alkaline HER activity of advanced Ni and Co-based electrocatalysts are also provided for comparison (Table R1). Table R1 has been added as Supplementary Table 7 in the Supplementary Information.

Table R1 HER activity of recent advanced electrocatalysts in 1 M KOH.

Catalysts	Overpotential at 10 mA cm ⁻²	Tafel slope	Reference
PS-MoNi@NF	30 mV	37 mV dec ⁻¹	Adv. Energy Mater. 2021, 11, 2003511
Ni-Mo-N/CFC	43 mV	70 mV dec ⁻¹	Nat. Commun. 2019, 10, 5335
Ni@C	27 mV	38 mV dec ⁻¹	Nano Lett. 2020, 20, 8375
Ru ₁ /D-NiFe LDH	18 mV	29 mV dec ⁻¹	Nat. Commun. 2021, 12, 4587
Mo-NiO/Ni	50 mV	86 mV dec ⁻¹	ACS Energy Lett. 2019, 4, 3002
F-Ni ₃ S ₄ /NF	29 mV	46 mV dec ⁻¹	Adv. Funct. Mater. 2021, 31, 2008578.
Ru@Ni-MOF	22 mV	40 mV dec ⁻¹	Angew. Chem. Int. Ed. 2021, 60, 22276
Pt-Ni NTA	23 mV	38 mV dec ⁻¹	Energy Environ. Sci. 2021, 14, 1594
MoO ₂ -FeP@C	103 mV	48 mV dec ⁻¹	Adv. Mater. 2020, 32, 2000455
Cu-Ni nanocages	140 mV	79 mV dec ⁻¹	ACS Catal. 2019, 9, 5084
Ni-CeF ₃ -VN	33 mV	37 mV dec ⁻¹	Adv. Funct. Mater. 2021, 31, 2104827
Pt _{SA} -NiO/Ni	26 mV	27 mV dec ⁻¹	Nat. Commun. 2021, 12, 3783
Cr-Ni NHs	75 mV	72 mV dec ⁻¹	J. Am. Chem. Soc. 2021, 143, 1399

Co ₁ /PCN	89 mV	52 mV dec ⁻¹	Nat. Catal. 2019, 2, 134
Ni ₅ P ₄ -Ru	54 mV	52.0 mV dec ⁻¹	Adv. Mater. 2020, 32, 1906972
FD-MoS ₂	164 mV	36 mV dec ⁻¹	Nat. Commun. 2022, 13, 2193
Pt ₁ /N-C	46 mV	37 mV dec ⁻¹	Nat. Commun. 2020, 11, 1029
NiRu _{0.13} -BDC	34 mV	32 mV dec ⁻¹	Nat. Commun. 2021, 12, 1369
Cs ₃ Rh ₂ I ₉ /NC-R	25 mV	30 mV dec ⁻¹	This work

3. I do not understand why Cs₃Rh₂I₉ got dispersed in rather polar DMF but can be quenched (precipitated) from the DMF with water. Authors, please provide explanation. More details and discussion should be provided here as it would allow for a better reproducibility and understanding of the experiments.

Response:

Thank very much for kind suggestions. The polar aprotic solvents, including dimethylformamide (DMF), dimethyl sulfoxide (DMSO), dimethylacetamide (DMAC) and N-methyl-2-pyrrolidone (NMP) are commonly used as solvents for halide perovskite (Nat. Sustain. 2021, 4, 277-285 and ACS Energy Lett. 2019, 4, 2983-2985). However, the solubility of halide perovskite in polar protic solvent (H₂O) is much lower due to the slow S_N2 reaction (Nat. Commun. 2016, 7, 11735 and Rev. Chem. Soc. 1962, 16, 163). To provide a better reproducibility and understanding of the experiment, we revised the main text and Methods as follows:

1. Cs₃Rh₂I₉ can be dissolved in N, N-dimethylformamide (DMF) to form a brownish yellow solution due to the polar aprotic property of DMF (Supplementary Fig. 6).
2. Subsequently, the above mixture was slowly added into 200 ml H₂O under fiercely

stirring. The induced polar protic solvent with slow S_N2 reaction kinetics results in the precipitation of $Cs_3Rh_2I_9$ ⁴⁷. The $Cs_3Rh_2I_9/NC$ was collected by suction filtration and washed by water and ethanol for several times.

4. Please clarify whether Rh 3d spectrum (bottom) recorded on the initial $Cs_3Rh_2I_9$ single crystals or non-immobilized $Cs_3Rh_2I_9$ (i.e. NC-free particles – called by authors “monocrystalline Rh”). If it is recorded on single crystals than the XPS of unsupported $Cs_3Rh_2I_9$ should be provided as it seems that support makes a significant difference and questions the assumption that grain boundaries play the key role in catalytic performance.

Response:

Thank you very much for your nice comments. The $Cs_3Rh_2I_9$ in Figure 2h (bottom) is obtained by the precipitation without the NC substrate. And the XPS results confirms the strong electronic interaction between $Cs_3Rh_2I_9$ clusters and NC in $Cs_3Rh_2I_9/NC$. The electronic interaction can not only decrease the formation energy of $Cs_3Rh_2I_9$ clusters to promote the uniform distribution (Adv. Energy Mater. 2021, 11, 2101050), but also influence the catalytic performance. To exclude the influence of electronic interaction in activity, we prepared Rh nanoparticle loaded on NC (Rh/NC) for comparison.

Figure R2 XPS spectra of Rh 3d for $Cs_3Rh_2I_9/NC-R$ and Rh/NC.

As shown in Figure R2, the binding energies of $Rh^{0}3d_{3/2}$ and $Rh^{0}3d_{5/2}$ for $Cs_3Rh_2I_9/NC-$

R are similar to those for Rh/NC, indicating the electronic state of Rh in Cs₃Rh₂I₉/NC-R and Rh/NC are the same. However, the HER activity of Cs₃Rh₂I₉/NC-R outperforms Rh/NC in both 1.0 M KOH and chlorine-alkali electrolyte due to the grain boundaries and tension stress in Cs₃Rh₂I₉/NC-R. Therefore, grain boundaries play the key role in catalytic performance.

5. I found the potential value of -0.035 V for the reduction wave in the initial CV scan too low given that Rh³⁺ ⇌ Rh standard reduction potential is in the order 0.6 V. To me it more consistent with surface oxidation. Same seems to apply to the NC-free Cs₃Rh₂I₉ as well as shown in the SI. Is there explanation for that?

Response:

Thanks very much for your excellent comments. As reviewer pointed, the standard reduction potentials of Rh²⁺ to Rh and Rh³⁺ to Rh are 0.6 and 0.758 V, respectively. And standard reduction potentials are obtained at 25 °C with ion concentration of 1 mol L⁻¹ (Rhⁿ⁺), which is different from the condition of Cs₃Rh₂I₉ reduction. According to the Pourbaix diagram, the experimental reduction potential is influenced by the pH, particle size, and reduction kinetics (Phys. Rev. B 2021, 85, 235438; Appl. Organometal. Chem. 2018, 32, e4118). Therefore, the reduction potential of Cs₃Rh₂I₉ (-0.035 V) is different from the standard reduction potentials of Rh³⁺ ions.

6. Please provide Raman spectrum recorded on pure Cs₃Rh₂I₉ which led “monocrystalline Rh”). The in operando spectra are very noisy and there is no peak assignment. I found it very, very strange that the reduction is almost instant especially at such low potential.

Response:

Figure R3 shows the in-situ Raman spectra of Cs₃Rh₂I₉/NC under the CV measurement. The peaks of Cs₃Rh₂I₉/NC at 142 and 124 cm⁻¹ correspond to the terminal Rh–I symmetric stretch in Cs₃Rh₂I₉. And we have replaced Supplementary Fig. 14 with Figure R3. Though Cs₃Rh₂I₉ is stable in air, it can be easily reduced under reduction

potential at only -0.035 V (vs. RHE).

Figure R3 The in-situ Raman spectra of Cs₃Rh₂I₉/NC under the CV measurement.

7. From the Fig. S20 it seems that unsupported Rh-particles are quite large, furthermore, plenty of grain boundaries are visible which seems to defy the hypothesis that improved characteristic of supported materials are due to grain boundaries. The particles are highly aggregated as well which given that button electrode is used makes it impossible to conclude whether it is the role of particles size, preferred orientation, support, or grain boundaries. This echoes the results presented in Fig. 4e and 4f which is based on two HRTEM images. Why are the authors so convinced that GB play dominant role? While monocrystalline NP are clearly 111 (Fig. 4f), I am not convinced from Fig4e and Fig.S25 that the Rh(111) is formed for supported NPs. Were there any SAED recorded to prove it? The bottom line is that the authors used a local method such as HRTEM (which may not be representative of the entire sample) to justify their hypothesis that grain boundaries play key role.

Response:

Thanks very much for your kind comments. The TEM of Cs₃Rh₂I₉-R shows massive aggregation of Rh particles. The accumulation of particles results in the appearance of dense boundaries in HRTEM which are not the GBs within the particle. To figure out its morphology, more HRTEM images are provided. As shown in Figure R4, the Rh particles in Cs₃Rh₂I₉-R are single crystals with average size of 4.3 ± 1.2 nm. The electrochemical activity was normalized to the ECSA to exclude the role of size.

Cs₃Rh₂I₉/NC-R enriched with GBs showed large current density of 0.065 mA cm⁻²_{ECSA} at the overpotential of 50 mV, much high than the 0.028 mA cm⁻²_{ECSA} of Cs₃Rh₂I₉-R, indicating its high intrinsic activity.

The SAED patterns of Cs₃Rh₂I₉/NC-R and Cs₃Rh₂I₉-R are shown in Figure R5. And the inner diffraction ring is composed of discrete points corresponding to the (111) plane of cubic Rh. And the Rh (111) plane in HRTEM was determined by the lattice fringe. And we selected multiple regions to confirm the GBs in Cs₃Rh₂I₉/NC-R. As shown in Figure R1, a sufficient number of HAADF-STEM images for Cs₃Rh₂I₉/NC-R confirm the abundant GBs within the nanoparticle.

Figure R4 HRTEM of Cs₃Rh₂I₉ after reduction (Cs₃Rh₂I₉-R). The inset in **b** shows particle size distribution.

Figure R5 **a** TEM image of $\text{Cs}_3\text{Rh}_2\text{I}_9/\text{NC-R}$. **b** SAED pattern of $\text{Cs}_3\text{Rh}_2\text{I}_9/\text{NC-R}$. **c** TEM image of $\text{Cs}_3\text{Rh}_2\text{I}_9\text{-R}$. **d** SAED pattern of $\text{Cs}_3\text{Rh}_2\text{I}_9\text{-R}$.

8. I wonder if authors prepared Rh-NP from RhCl_3 (mentioned in the methods) why did not they used them as a benchmark? As mentioned above Pt/C is not a useful benchmark here as it is not used in commercial alkaline electrolysers and it would be more useful to compare between Rh-NP prepared by traditional method (RhCl_3) and the one proposed by the authors ($\text{Cs}_3\text{Rh}_2\text{I}_9$). This is more important as the article is interesting from fundamental science of view (I doubt solid state method for synthesis of $\text{Cs}_3\text{Rh}_2\text{I}_9$ is practical to be adopted). Therefore, it would be good to benchmark against RuCl_3 -made NPs whether immobilised on NC or not. I understand that the authors showed in Fig. 5b the results from other groups but such results may be influenced by numerous factors such as the nature of support which seems to play substantial role.

Just in case that it is lost in the text above authors please add the results of

electrochemical and TEM assessment of NP prepared by reduction of RhCl₃ with sodium borohydride.

Overall, I disagree with authors that “The enhanced activity may be ascribed to the polycrystalline Rh with tensile stress” the evidence presented in their work seem to point to the role of the substrate (which improved dispersibility and high active site exposure) and/or size of nanoparticles. Still, I believe this is a solid and well-presented manuscript and the article is suitable for Nature Comms. When the technical points are addressed.

Response:

Thanks very much for your suggestions. The TEM images of Rh-NP from RhCl₃ (Rh/NC) are shown in Supplementary Figure 22, and its electrochemical performance has updated in the main text. The Rh particles in Rh/NC uniformly distributes on the NC substrate. And the electronic state of Rh in and Rh/NC is consistent with that in Cs₃Rh₂I₉/NC-R (Figure R2). Subsequently, the electrochemical performance in both 1.0 M KOH and chlorine-alkali electrolyte are compared. As shown in Figure R6a, the overpotential at 10 mA cm⁻² for Cs₃Rh₂I₉/NC-R, Cs₃Rh₂I₉-R and Rh/NC are 25, 123 and 31 mV, respectively. The Tafel slope of Cs₃Rh₂I₉/NC-R is only 30.3 mV dec⁻¹, lower than that of Rh/NC (35.5 mV dec⁻¹). And Cs₃Rh₂I₉/NC-R showed large current density of 0.065 mA cm⁻²_{ECSA} at the overpotential of 50 mV, much high than the 0.028 mA cm⁻²_{ECSA} of Cs₃Rh₂I₉-R, indicating its high intrinsic activity. In the simulated chlorine-alkali electrolyte, the overpotentials for Cs₃Rh₂I₉/NC-R are 21, 65, and 107 mV to reach current densities of 10, 50 and 100 mA cm⁻², respectively, significantly lower than those of Cs₃Rh₂I₉-R and Rh/NC (Figure R6c). The mass activity of Cs₃Rh₂I₉/NC-R is 782.8 mA mg⁻¹_{Rh} at -50 mV vs. RHE, outperforming Cs₃Rh₂I₉-R (22.1 mA mg⁻¹_{Rh}) and Rh/NC (175.9 mA mg⁻¹_{Rh}). These results confirm the excellent alkaline HER activity of Cs₃Rh₂I₉/NC-R.

Figure R6 HER activity in 1.0 M KOH and chlor-alkali electrolyte. **a** LSV curves of all samples in 1.0 M KOH. **b** Comparison of overpotential at 10 mA cm^{-2} and Tafel slope for various Rh-based catalysts in 1.0 M KOH. **c** LSV curves of all samples in chlorine-alkali electrolyte. **d** Mass activity normalized to the mass of Rh/Pt. **e** Comparison of mass activity and specific activity at the overpotential of 50 mV in chlorine-alkali electrolyte. **f** Stability of $\text{Cs}_3\text{Rh}_2\text{I}_9/\text{NC-R}$ at 10 mA cm^{-2} in chlorine-alkali electrolyte.

Through the comparison of current density and normalized activity for $\text{Cs}_3\text{Rh}_2\text{I}_9/\text{NC-R}$, $\text{Cs}_3\text{Rh}_2\text{I}_9\text{-R}$ and Rh/NC, it is reasonable to exclude the influence of particle size and substrate on HER performance and thereby confirming the role of GB in $\text{Cs}_3\text{Rh}_2\text{I}_9/\text{NC-R}$.

R.

Moreover, the enhanced water dissociation ability in Cs₃Rh₂I₉/NC-R was verified by in situ Raman spectra. Figure R7a shows obvious interfacial water on Cs₃Rh₂I₉/NC-R surface in 1M KOH solution over the potential range from 0 to -0.3 V (vs. RHE). The peaks at 1588 and 1632 cm⁻¹ correspond to the G band of carbon substrate and the adsorbed water on Rh, respectively (Nature, 1965, 205, 170, J. Electroanal. Chem., 1996, 415, 175-178). As the potential decreased, the intensity of H-O-H bending increases sharply. However, such phenomenon is absent in Rh/NC (Figure R7b). The strong H-O-H bending sign on twined Rh with abundant grain boundaries suggest the high water dissociation kinetics on Cs₃Rh₂I₉/NC-R during electrocatalysis.

Figure R7 In situ Raman spectra of interfacial water on Cs₃Rh₂I₉/NC-R (a) and Rh/NC (b).

And the DFT calculation was also performed to confirm the role of GB and tensile stress in HER. In alkaline HER, the water dissociation, as a pre-reaction to form an adsorbed proton, plays a more critical role in electrocatalysis. As shown in Supplementary Fig. 38, the increase of H-O-H angle and water molecule binding energy agrees well with degree of distortion and crystal tension, indicating the enhanced ability of water dissociation. Therefore, GBs are proved to play the key role in enhanced the hydrogen evolution activity.

Reviewer #3 (Remarks to the Author):

The authors reported the Rh nanoparticles with abundant GBs, and detailly elucidated the synthetic process and analyzed the formed mechanism of GB. In particular, the top-down and bottom-up methods for Rh nanoparticles preparation were compared by many physical characterization techniques. As expected, the GB-enriched Rh nanoparticles demonstrated the excellent catalytic activity, which is superior to monocrystalline Rh without GBs. It claims that such activity results from the boosted water dissociation kinetics by the GB with strain stress. However, there are many critical issues (see following) that prevent me from recommending this paper to publish in the high-quality journal.

1. The authors claim that the GB-enriched Rh nanoparticles were prepared by the bottom-up methods based on the NC used as support. Although many reported literatures have similar description, namely, there is a substrate as a nucleation site that is considered as bottom-up methods. However, the specific process of electrochemical reduction of Cs₃Rh₂I₉ perovskite clusters is no basically comprehensive research.

Response:

Thanks very much for your positive comments. We have tried to observe the reduction process of Cs₃Rh₂I₉ perovskite clusters by TEM, whereas the clusters are unstable under the high-energy measurement (Figure R8). The average size of Cs₃Rh₂I₉ clusters is only 1.7 nm. However, the size of Rh are 2.2 nm after reduction due to the aggregation of smaller Rh clusters during the electrochemical reduction process. To elucidate the process of reconstruction, the reduction was conducted by the potentiostatic measurement at -0.03 V vs. RHE. The in-situ electrochemical Quartz Crystal Microbalance (EQCM) experiment, ex-situ Inductively Coupled Plasma Mass Spectrometry (ICP-MS) and XPS spectra characterizations were carried out to elucidate the Cs and I extraction and Rh reduction during the electrochemical reduction.

Figure R8 **a** HAADF-STEM of $\text{Cs}_3\text{Rh}_2\text{I}_9/\text{NC}$ and **b** decomposed $\text{Cs}_3\text{Rh}_2\text{I}_9/\text{NC}$.

The in-situ EQCM measurement indicates that the quality of the electrode continuously decreases and then stabilizes after about 5 min (Figure R9a). The ICP-MS results show the content of Cs^+ ions in the electrolyte continuously increases in the initial 5 min while Rh in the electrolyte was barely detected (Figure R9b). The ex-situ XPS spectra verify the content of Rh^{3+} decreases while the content of Rh^0 increases with the reduction time (Figure R9c). Therefore, the Cs and I gradually extracted from the perovskite clusters and Rh^{3+} was reduced to Rh^0 during the reconstruction. Meanwhile, the small Rh clusters combined with neighboring clusters into particles (~ 2.2 nm) to decrease the surface energy (Figure R10).

Figure R9 **a** Mass change of the $\text{Cs}_3\text{Rh}_2\text{I}_9/\text{NC}$ electrode monitored by in-situ EQCM experiment. **b** ICP-MS of the Cs and Rh contents in the electrolyte at different reduction time. **c** XPS spectra of Rh 3d for $\text{Cs}_3\text{Rh}_2\text{I}_9/\text{NC}$ at different reduction time.

Figure R10 HAADF-STEM images of adjacent Rh particles in reduced $\text{Cs}_3\text{Rh}_2\text{I}_9/\text{NC}$ ($\text{Cs}_3\text{Rh}_2\text{I}_9/\text{NC-R}$).

2. There is no experimental investigation to verify the enhanced water dissociation kinetics. The DFT calculations is only the theory.

Response:

Thanks very much for Reviewer's suggestion. In situ Raman was taken to prove the enhanced water dissociation kinetics of the catalyst. Figure R11a shows obvious interfacial water on $\text{Cs}_3\text{Rh}_2\text{I}_9/\text{NC-R}$ surface in 1M KOH solution over the potential range from 0 to -0.3 V (vs. RHE). The peaks at 1588 and 1632 cm^{-1} correspond to the G band of carbon substrate and the adsorbed water on Rh, respectively (Nature, 1965, 205, 170, J. Electroanal. Chem., 1996, 415, 175-178). As the potential decreased, the intensity of H-O-H bending increases sharply. However, such phenomenon is absent in Rh/NC (Figure R11b). The strong H-O-H bending sign on twined Rh with abundant grain boundaries suggest the high water dissociation kinetics on $\text{Cs}_3\text{Rh}_2\text{I}_9/\text{NC-R}$ during electrocatalysis. This result is consistent with the DFT calculations.

Figure R11 In situ Raman spectra of interfacial water on Cs₃Rh₂I₉/NC-R (a) and Rh/NC (b).

We have added Figure R11 as new Supplementary Fig. 39 in Supplementary Information. The corresponding discussion was also added in the main text:

The enhanced water dissociation ability in Cs₃Rh₂I₉/NC-R was further verified by in situ Raman spectra. Supplementary Fig. 39a shows obvious interfacial water on Cs₃Rh₂I₉/NC-R surface in 1M KOH solution over the potential range from 0 to -0.3 V (vs. RHE). The peaks at 1588 and 1632 cm⁻¹ correspond to the G band of carbon substrate and the adsorbed water on Rh, respectively^{45, 46}. As the potential decreased, the intensity of H-O-H bending increases sharply. However, such phenomenon is absent in Rh/NC without GBs (Supplementary Fig. 39b).

3. According to the XPS and EDX with limited detection ability, it is not certain that there is no Cs and I elements in Rh nanoparticles. Thus, the HER activity origin needs to be further investigated because the synergistic effect between elements is very important in enhancing HER performance.

Response:

Thanks very much for Reviewer's suggestion. As the synergistic effect between elements is very important in enhancing HER performance, it is necessary to confirm the contents of Cs and I in Rh nanoparticles. ICP-MS results show the mass ratio of Rh, Cs and I is 1:0.0084:0.0024 in Cs₃Rh₂I₉/NC-R, indicating the product was reduced to Rh

particles. The small amount of Cs and I maybe ascribed to the adsorbed Cs and I ions on the NC substrate. The ICP-MS results have been added in the Supplementary Table 5.

4. In term of DFT calculations, especial the constructed atom model, authors did not consider the role of NC in HER process, but both XPS and XAS results evidenced the strong electronic interactions between Rh and NC.

Response:

As reviewer pointed, both XPS and XAS results evidenced the strong electronic interactions between twined Rh particles and NC substrate. As the electronic interactions may influence the catalytic performance, it is necessary to set up reference sample to rule out the role of NC in the HER process. Rh/NC prepared by chemical reduction of RhCl₃ with sodium borohydride was served as a benchmark. XPS results show binding energies of Rh⁰3d_{3/2} and Rh⁰3d_{5/2} for Rh/NC are similar to those for Cs₃Rh₂I₉/NC-R (Figure R12), indicating the electronic state of Rh in Rh/NC and Cs₃Rh₂I₉/NC-R are the same. However, the HER activity of Cs₃Rh₂I₉/NC-R outperforms Rh/NC in both 1.0 M KOH and chlorine-alkali electrolyte due to the grain boundaries and tension stress in Cs₃Rh₂I₉/NC-R. Therefore, grain boundaries play the key role in catalytic performance. Figure R12 was added as Supplementary Fig. 22c in Supplementary Information.

Figure R12 XPS spectra of Rh 3d for Cs₃Rh₂I₉/NC, Cs₃Rh₂I₉/NC-R and Rh/NC.

5. In experimental method, there is a mistake. The RuCl₃ should be substituted with RhCl₃ in “synthesis of Rh/NC”.

Response:

Thanks very much for Reviewer’s careful check of our manuscript. The mistake has been corrected in the revised main text.

Reviewer #1 (Remarks to the Author):

This manuscript deals with the preparation of Rh nanoparticles containing grain boundaries (GBs) via the electrochemical reduction of perovskite $\text{Cs}_3\text{Rh}_2\text{I}_9$. A major claim by the authors is that this reduction process has an advantage of obtaining defective Rh nanoparticles with GBs and their HER activity can be significantly enhanced by the presence of GBs. Although the notable improvement of catalytic activity by an order of magnitude in HER current density is highly appreciated, the following issues prevent us from recommending publication of this manuscript.

1. Even though the provision of clear-cut evidence proving the presence of GBs in Rh nanoparticles is of great significance in this study, no reliable images for GBs are provided in the manuscript. In particular, the authors repeatedly show exactly the same HAADF image in both Figure 1b and Figure 4e. Unless a sufficient number of clear images for GBs are given to confirm the existence of GBs in the nanoparticles, very few readers would agree with the authors' claim (for example, see the previous report, ACS Catal. 10, 12575-12581 (2020)).

Response: We appreciate your comments very much. We have been added enough HAADF-STEM images for $\text{Cs}_3\text{Rh}_2\text{I}_9/\text{NC-R}$ enriched with GBs (Figure R1) as Supplementary Fig. 25 in the Supplementary Information. All these images provide strong evidence that the GBs are present in lots of Rh nanoparticles for the $\text{Cs}_3\text{Rh}_2\text{I}_9/\text{NC-R}$ sample. Figure 4e in the main text was replaced by Figure R1a.

Figure R1 HAADF-STEM images of $\text{Cs}_3\text{Rh}_2\text{I}_9/\text{NC-R}$.

2. Furthermore, the GBs in Figure 4e do not really appear to be grain boundaries. They are rather very likely to be stacking faults. In FCC/HCP metals, such stacking faults frequently form during nanoparticle synthesis at low temperature (see ref. 36 in this manuscript). It is thus necessary to clarify whether the Rh nanoparticles contain GBs or not.

Response:

Thank you very much for your careful check of our manuscript. In this work, the Rh particles are formed by in-situ electrochemical reduction of $\text{Cs}_3\text{Rh}_2\text{I}_9$ perovskite clusters. The average size of initial $\text{Cs}_3\text{Rh}_2\text{I}_9$ clusters is only 1.7 nm. However, the average size of Rh particles increased to 2.2 nm after reduction due to the coupling of in-situ-formed ultra-small Rh clusters with high surface energy, which could lead to the formation of twin grain boundaries. Besides, lots of STEM images show the obvious angles of lattice planes in the twined particles, suggesting the Rh nanoparticles contain grain boundaries.

3. Although the presence of GBs has been already reported to be beneficial in CO₂ reduction electrocatalysis, no direct correlation between the GBs and the enhancement of the HER activity is shown by the DFT calculations in this study. Readers would wonder why the tensile strain induce by GBs in Rh can significantly improve the HER activity. Moreover, the calculation on the binding energy of water molecule in Supplementary Fig. 39 does not appear to be logically reasonable to explain the role of GBs. From this viewpoint of the authors, if we could obtain any catalyst with the highest binding energy, it would show the best HER activity, which seriously contradicts the well-known ‘Sabatire principle’. Even though the effect of tensile strain has been discussed in Ag (ref. 36 in this manuscript), more rigorous and systematic calculations should be carried out in Rh for readers.

Response:

We sincerely agree with you that a common understanding of adsorption strength follows the “Sabatire principle”. However, in HER, water is used as reactant, instead of intermediate. Therefore, the selection of “moderate adsorption” of water as the most appropriate adsorption strength toward alkaline HER is not appropriate. Besides, as reported in previous study, the adsorption energy and dissociative kinetic barrier of H₂O have a linear Brønsted-Evans-Polanyi (BEP) relationship (Nat. Chem. 2009, 1, 37; Nat. Commun. 2022, 13, 1270). Thus, the adsorption energy of H₂O can be used as an activity descriptor for the kinetic barrier of water dissociation. The higher H₂O binding energy is, the lower the barrier for H₂O dissociation will be. In this work, the binding energy of water molecule on Rh is 0.27 eV, and the value increases to 0.33 eV when water adsorbs on Rh with 3% tension (Supplementary Fig. 39), indicating the enhanced water dissociation ability of Rh with strain. In the structural models of adsorbed water molecule on Rh, the largest H–O–H angle of 105.25° is found when water molecule is adsorbed on GBs, which further shows GBs acting as sites would activate the H–OH bond. To make it easier for readers to understand, more discussion has been added in the main text:

The linear Brønsted-Evans-Polanyi relationship between adsorption energy and dissociative kinetic barrier of H₂O allows the use of binding energy of water molecule as the activity descriptor for alkaline HER. The sites at GBs and tensed Rh atoms show high H₂O binding energy. The enhanced water dissociation ability in Cs₃Rh₂I₉/NC-R was further verified by in situ Raman spectra. Supplementary Fig. 40a shows obvious interfacial water on Cs₃Rh₂I₉/NC-R surface in 1M KOH solution over the potential range from 0 to -0.3 V (vs. RHE). The peaks at 1588 and 1632 cm⁻¹ correspond to the G band of carbon substrate and the adsorbed water on Rh, respectively^{45, 46}. As the potential decreased, the intensity of H-O-H bending increases sharply. However, such phenomenon is absent in Rh/NC without GBs (Supplementary Fig. 40b). Thus, the abundant and multiply active sites on twin crystal Rh can be effective for cleaving the H-OH bond, contributing to the high electrochemical performance toward HER under alkaline condition.

4. It should be noted that nitrogen-doped carbon (NC) was used as a support material in this study. Consequently, there is concern about the formation of single-atom Rh during the electrochemical reduction of perovskite Cs₃Rh₂I₉. Although single-atom Rh is not directly captured in the HAADF image Figure 4a, it is worthwhile to examine its presence on the NC support for clarification.

Response:

Thank you very much for the valuable comments. The extended X-ray absorption fine structure (EXAFS) is sensitive to chemical and coordination states of element and can provide coordination number and average bond length between single atom and nearby atoms. As shown in Supplementary Fig. 24, Fourier transform EXAFS of Rh K-edge in R-space for Cs₃Rh₂I₉/NC-R exhibits the shell at 2.45 Å, similar to that of Rh foil. Fitting result proves the shell corresponds to the Rh-Rh scattering path with coordination number of 8.0. And no Rh-N(C) scattering path was detected. Besides, single-atom Rh is not directly captured in the HAADF image. Therefore, only Rh particles are formed in Cs₃Rh₂I₉/NC-R.

Reviewer #2 (Remarks to the Author):

The paper reports the synthesis, characterisation, and comprehensive electrochemical characterisation of free-standing and immobilised on carbon-related support Rh-nanoparticles. The authors proposed in situ electrochemical reduction of Cs₃Rh₂I₉ precursor as a general method which seems to help them achieve a good distribution of Rh nanoparticles on the support. This is a good proposal despite solid-state synthesis Cs₃Rh₂I₉ is highly complicated and few labs (and even less industrial users) are equipped with ampoule sealing apparatus. Still from understanding the role of surface passivation by surfactant the method has its merits (keeping in mind that the cost of CsI / I is negligible compared with the cost of Rh). The authors benchmarked the NP prepared by the proposed approach with Pt/C but seem to neglect the materials from a traditional method for the synthesis of Rh-nanoparticles via RhCl₃ route. Overall, this is a very consistent work and I like the thorough approach authors took to support their results.

Response:

Thanks very much for your positive comments and support on our work. We have revised the manuscript according to your valuable comments. Detailed discussions are listed in the following point-by-point response. Hopefully the revised edition could satisfy you.

1. However, the authors try to make the case that grain boundaries play the key role in improving the catalytic performance of nanoparticles (when compared with unsupported nanoparticles). This is something I was not convinced completely. I am not sure why authors decided to go a rather treacherous route and insist that grain boundaries are the reason for improved performance given that the work is quite consistent and would merit publication without this speculative (and hard to prove) idea. Well, since the authors are convinced that grain boundaries are essential, I would like them to provide more convincing evidence and thus, there are some additional experiments I would like them to conduct: the materials should be benchmarked against

Rh-NP prepared by reduction of RhCl₃ with sodium borohydride.

Still, despite the contested issue of the catalytic properties originating from grain boundaries I recommend publication of this article after technical issues are addressed:

Response:

Thanks for Reviewer's comprehensive suggestion to this work. In this work, we developed a bottom-up evolution route to prepare twinned Rh nanoparticles with high-density and exposed grain boundaries (GBs) *via* in-situ electrochemical reduction of Cs₃Rh₂I₉ perovskite clusters. However, in the initial stage of this work, we only tried to evaluate the HER activity of Cs₃Rh₂I₉/NC. The reduction peak in the first CV curve and increased particle size of loaded nanoparticle after reduction aroused our curiosity. Therefore, in-situ electrochemical Quartz Crystal Microbalance experiment, Inductively Coupled Plasma Mass Spectrometry and X-ray photoelectron spectroscopy were taken to elucidate the process of the electrochemical reduction. And high angle annular dark-field scanning transmission electron microscopy showed obvious GBs in reduced Cs₃Rh₂I₉/NC (Cs₃Rh₂I₉/NC-R).

Nanocrystals with twinned structure are considered as advanced catalysts due to stressed grain and boundaries from surface mismatches. The existing structural perturbation could lead to the optimization of the electronic state, the adsorption characteristics and the reaction kinetics of the catalyst. For examples, a GB-rich metallic copper showed much higher electrochemical carbon dioxide reduction activity than the copper without GB due to the enhanced CO binding and C–C coupling properties (J. Am. Chem. Soc. 2020, 142, 6878); the catalytic rate was positively correlated with GB density in Pd catalyst (Science 2021, 373, 1518) and the GBs in monolayer molybdenum disulphide acted as new highly active and tunable catalytic sites for HER (Nat. Commun. 2019, 10, 1348). Therefore, when we found the high-density and exposed GBs in Cs₃Rh₂I₉/NC and its high alkaline HER performance, we tried to prove that the GB are responsible for the enhanced activity. In 1.0 M KOH, the GB-rich Cs₃Rh₂I₉/NC-R exhibited high mass activity of 846 mA mg⁻¹_{Rh} at the overpotential of 50 mV, which is about 22.3 times that of GB-free Cs₃Rh₂I₉-R. To

compare their intrinsic activity, the current was also normalized to the electrochemical active surface area (ECSA). Cs₃Rh₂I₉/NC-R showed large current density of 0.065 mA cm⁻²_{ECSA} at the overpotential of 50 mV, which is 30% higher than that of Rh/NC without GB. Besides, DFT calculation confirmed GBs can act as sites to activate the H–OH bond, which plays a critical role in boosting alkaline hydrogen evolution.

As reviewer pointed, Rh/NC prepared by chemical reduction of RhCl₃ with sodium borohydride should serve as a benchmark. The electrochemical performance of Rh/NC in both 1.0 M KOH and chlorine-alkali electrolyte have been added in the main text. Combined theoretical calculations with previous report, GBs are proved to play the key role in enhanced the electrolysis.

2. The authors try to give the impression to the reader that Pt/C is the catalyst of choice for the HER in alkaline electrolyzers. This is not true, it is a key catalyst in PEM electrolyzers, i.e. acidic electrolytes. In alkaline conditions, Ni-Mo or Ni-Fe-Mo alloys even Ni (foam) are widely used in commercial electrolyzers. This is because the limiting stage is OER and the activity of HER does not matter that much on overall performance of the stack. I would like to the authors to soften their statement a little bit and mention some good work on alkaline electrolyzers. The notion that Pt/C is used in alkaline electrolysis is not helpful as it propagates the erroneous perception and confuses unexperienced readers. This is something that should be avoided given general readership of Nature Comms.

Response:

Thanks very much for your valuable suggestion. To avoid the misunderstandings for readers, we have softened our statement about Pt/C, and focused more on the comparison of Cs₃Rh₂I₉/NC-R to Cs₃Rh₂I₉-R and Rh/NC. Besides, the alkaline HER activity of advanced Ni and Co-based electrocatalysts are also provided for comparison (Table R1). Table R1 has been added as Supplementary Table 7 in the Supplementary Information.

Table R1 HER activity of recent advanced electrocatalysts in 1 M KOH.

Catalysts	Overpotential at 10 mA cm ⁻²	Tafel slope	Reference
PS-MoNi@NF	30 mV	37 mV dec ⁻¹	Adv. Energy Mater. 2021, 11, 2003511
Ni-Mo-N/CFC	43 mV	70 mV dec ⁻¹	Nat. Commun. 2019, 10, 5335
Ni@C	27 mV	38 mV dec ⁻¹	Nano Lett. 2020, 20, 8375
Ru ₁ /D-NiFe LDH	18 mV	29 mV dec ⁻¹	Nat. Commun. 2021, 12, 4587
Mo-NiO/Ni	50 mV	86 mV dec ⁻¹	ACS Energy Lett. 2019, 4, 3002
F-Ni ₃ S ₄ /NF	29 mV	46 mV dec ⁻¹	Adv. Funct. Mater. 2021, 31, 2008578.
Ru@Ni-MOF	22 mV	40 mV dec ⁻¹	Angew. Chem. Int. Ed. 2021, 60, 22276
Pt-Ni NTA	23 mV	38 mV dec ⁻¹	Energy Environ. Sci. 2021, 14, 1594
MoO ₂ -FeP@C	103 mV	48 mV dec ⁻¹	Adv. Mater. 2020, 32, 2000455
Cu-Ni nanocages	140 mV	79 mV dec ⁻¹	ACS Catal. 2019, 9, 5084
Ni-CeF ₃ -VN	33 mV	37 mV dec ⁻¹	Adv. Funct. Mater. 2021, 31, 2104827
Pt _{SA} -NiO/Ni	26 mV	27 mV dec ⁻¹	Nat. Commun. 2021, 12, 3783
Cr-Ni NHs	75 mV	72 mV dec ⁻¹	J. Am. Chem. Soc. 2021, 143, 1399

Co ₁ /PCN	89 mV	52 mV dec ⁻¹	Nat. Catal. 2019, 2, 134
Ni ₅ P ₄ -Ru	54 mV	52.0 mV dec ⁻¹	Adv. Mater. 2020, 32, 1906972
FD-MoS ₂	164 mV	36 mV dec ⁻¹	Nat. Commun. 2022, 13, 2193
Pt ₁ /N-C	46 mV	37 mV dec ⁻¹	Nat. Commun. 2020, 11, 1029
NiRu _{0.13} -BDC	34 mV	32 mV dec ⁻¹	Nat. Commun. 2021, 12, 1369
Cs ₃ Rh ₂ I ₉ /NC-R	25 mV	30 mV dec ⁻¹	This work

3. I do not understand why Cs₃Rh₂I₉ got dispersed in rather polar DMF but can be quenched (precipitated) from the DMF with water. Authors, please provide explanation. More details and discussion should be provided here as it would allow for a better reproducibility and understanding of the experiments.

Response:

Thank very much for kind suggestions. The polar aprotic solvents, including dimethylformamide (DMF), dimethyl sulfoxide (DMSO), dimethylacetamide (DMAC) and N-methyl-2-pyrrolidone (NMP) are commonly used as solvents for halide perovskite (Nat. Sustain. 2021, 4, 277-285 and ACS Energy Lett. 2019, 4, 2983-2985). However, the solubility of halide perovskite in polar protic solvent (H₂O) is much lower due to the slow S_N2 reaction (Nat. Commun. 2016, 7, 11735 and Rev. Chem. Soc. 1962, 16, 163). To provide a better reproducibility and understanding of the experiment, we revised the main text and Methods as follows:

1. Cs₃Rh₂I₉ can be dissolved in N, N-dimethylformamide (DMF) to form a brownish yellow solution due to the polar aprotic property of DMF (Supplementary Fig. 6).
2. Subsequently, the above mixture was slowly added into 200 ml H₂O under fiercely

stirring. The induced polar protic solvent with slow S_N2 reaction kinetics results in the precipitation of $Cs_3Rh_2I_9$ ⁴⁷. The $Cs_3Rh_2I_9/NC$ was collected by suction filtration and washed by water and ethanol for several times.

4. Please clarify whether Rh 3d spectrum (bottom) recorded on the initial $Cs_3Rh_2I_9$ single crystals or non-immobilized $Cs_3Rh_2I_9$ (i.e. NC-free particles – called by authors “monocrystalline Rh”). If it is recorded on single crystals than the XPS of unsupported $Cs_3Rh_2I_9$ should be provided as it seems that support makes a significant difference and questions the assumption that grain boundaries play the key role in catalytic performance.

Response:

Thank you very much for your nice comments. The $Cs_3Rh_2I_9$ in Figure 2h (bottom) is obtained by the precipitation without the NC substrate. And the XPS results confirms the strong electronic interaction between $Cs_3Rh_2I_9$ clusters and NC in $Cs_3Rh_2I_9/NC$. The electronic interaction can not only decrease the formation energy of $Cs_3Rh_2I_9$ clusters to promote the uniform distribution (Adv. Energy Mater. 2021, 11, 2101050), but also influence the catalytic performance. To exclude the influence of electronic interaction in activity, we prepared Rh nanoparticle loaded on NC (Rh/NC) for comparison.

Figure R2 XPS spectra of Rh 3d for $Cs_3Rh_2I_9/NC-R$ and Rh/NC.

As shown in Figure R2, the binding energies of $Rh^{0}3d_{3/2}$ and $Rh^{0}3d_{5/2}$ for $Cs_3Rh_2I_9/NC-$

R are similar to those for Rh/NC, indicating the electronic state of Rh in Cs₃Rh₂I₉/NC-R and Rh/NC are the same. However, the HER activity of Cs₃Rh₂I₉/NC-R outperforms Rh/NC in both 1.0 M KOH and chlorine-alkali electrolyte due to the grain boundaries and tension stress in Cs₃Rh₂I₉/NC-R. Therefore, grain boundaries play the key role in catalytic performance.

5. I found the potential value of -0.035 V for the reduction wave in the initial CV scan too low given that Rh³⁺ ⇌ Rh standard reduction potential is in the order 0.6 V. To me it more consistent with surface oxidation. Same seems to apply to the NC-free Cs₃Rh₂I₉ as well as shown in the SI. Is there explanation for that?

Response:

Thanks very much for your excellent comments. As reviewer pointed, the standard reduction potentials of Rh²⁺ to Rh and Rh³⁺ to Rh are 0.6 and 0.758 V, respectively. And standard reduction potentials are obtained at 25 °C with ion concentration of 1 mol L⁻¹ (Rhⁿ⁺), which is different from the condition of Cs₃Rh₂I₉ reduction. According to the Pourbaix diagram, the experimental reduction potential is influenced by the pH, particle size, and reduction kinetics (Phys. Rev. B 2021, 85, 235438; Appl. Organometal. Chem. 2018, 32, e4118). Therefore, the reduction potential of Cs₃Rh₂I₉ (-0.035 V) is different from the standard reduction potentials of Rh³⁺ ions.

6. Please provide Raman spectrum recorded on pure Cs₃Rh₂I₉ which led “monocrystalline Rh”). The in operando spectra are very noisy and there is no peak assignment. I found it very, very strange that the reduction is almost instant especially at such low potential.

Response:

Figure R3 shows the in-situ Raman spectra of Cs₃Rh₂I₉/NC under the CV measurement. The peaks of Cs₃Rh₂I₉/NC at 142 and 124 cm⁻¹ correspond to the terminal Rh–I symmetric stretch in Cs₃Rh₂I₉. And we have replaced Supplementary Fig. 14 with Figure R3. Though Cs₃Rh₂I₉ is stable in air, it can be easily reduced under reduction

potential at only -0.035 V (vs. RHE).

Figure R3 The in-situ Raman spectra of Cs₃Rh₂I₉/NC under the CV measurement.

7. From the Fig. S20 it seems that unsupported Rh-particles are quite large, furthermore, plenty of grain boundaries are visible which seems to defy the hypothesis that improved characteristic of supported materials are due to grain boundaries. The particles are highly aggregated as well which given that button electrode is used makes it impossible to conclude whether it is the role of particles size, preferred orientation, support, or grain boundaries. This echoes the results presented in Fig. 4e and 4f which is based on two HRTEM images. Why are the authors so convinced that GB play dominant role? While monocrystalline NP are clearly 111 (Fig. 4f), I am not convinced from Fig4e and Fig.S25 that the Rh(111) is formed for supported NPs. Were there any SAED recorded to prove it? The bottom line is that the authors used a local method such as HRTEM (which may not be representative of the entire sample) to justify their hypothesis that grain boundaries play key role.

Response:

Thanks very much for your kind comments. The TEM of Cs₃Rh₂I₉-R shows massive aggregation of Rh particles. The accumulation of particles results in the appearance of dense boundaries in HRTEM which are not the GBs within the particle. To figure out its morphology, more HRTEM images are provided. As shown in Figure R4, the Rh particles in Cs₃Rh₂I₉-R are single crystals with average size of 4.3 ± 1.2 nm. The electrochemical activity was normalized to the ECSA to exclude the role of size.

Cs₃Rh₂I₉/NC-R enriched with GBs showed large current density of 0.065 mA cm⁻²_{ECSA} at the overpotential of 50 mV, much high than the 0.028 mA cm⁻²_{ECSA} of Cs₃Rh₂I₉-R, indicating its high intrinsic activity.

The SAED patterns of Cs₃Rh₂I₉/NC-R and Cs₃Rh₂I₉-R are shown in Figure R5. And the inner diffraction ring is composed of discrete points corresponding to the (111) plane of cubic Rh. And the Rh (111) plane in HRTEM was determined by the lattice fringe. And we selected multiple regions to confirm the GBs in Cs₃Rh₂I₉/NC-R. As shown in Figure R1, a sufficient number of HAADF-STEM images for Cs₃Rh₂I₉/NC-R confirm the abundant GBs within the nanoparticle.

Figure R4 HRTEM of Cs₃Rh₂I₉ after reduction (Cs₃Rh₂I₉-R). The inset in **b** shows particle size distribution.

Figure R5 **a** TEM image of $\text{Cs}_3\text{Rh}_2\text{I}_9/\text{NC-R}$. **b** SAED pattern of $\text{Cs}_3\text{Rh}_2\text{I}_9/\text{NC-R}$. **c** TEM image of $\text{Cs}_3\text{Rh}_2\text{I}_9\text{-R}$. **d** SAED pattern of $\text{Cs}_3\text{Rh}_2\text{I}_9\text{-R}$.

8. I wonder if authors prepared Rh-NP from RhCl_3 (mentioned in the methods) why did not they used them as a benchmark? As mentioned above Pt/C is not a useful benchmark here as it is not used in commercial alkaline electrolysers and it would be more useful to compare between Rh-NP prepared by traditional method (RhCl_3) and the one proposed by the authors ($\text{Cs}_3\text{Rh}_2\text{I}_9$). This is more important as the article is interesting from fundamental science of view (I doubt solid state method for synthesis of $\text{Cs}_3\text{Rh}_2\text{I}_9$ is practical to be adopted). Therefore, it would be good to benchmark against RuCl_3 -made NPs whether immobilised on NC or not. I understand that the authors showed in Fig. 5b the results from other groups but such results may be influenced by numerous factors such as the nature of support which seems to play substantial role.

Just in case that it is lost in the text above authors please add the results of

electrochemical and TEM assessment of NP prepared by reduction of RhCl₃ with sodium borohydride.

Overall, I disagree with authors that “The enhanced activity may be ascribed to the polycrystalline Rh with tensile stress” the evidence presented in their work seem to point to the role of the substrate (which improved dispersibility and high active site exposure) and/or size of nanoparticles. Still, I believe this is a solid and well-presented manuscript and the article is suitable for Nature Comms. When the technical points are addressed.

Response:

Thanks very much for your suggestions. The TEM images of Rh-NP from RhCl₃ (Rh/NC) are shown in Supplementary Figure 22, and its electrochemical performance has updated in the main text. The Rh particles in Rh/NC uniformly distributes on the NC substrate. And the electronic state of Rh in and Rh/NC is consistent with that in Cs₃Rh₂I₉/NC-R (Figure R2). Subsequently, the electrochemical performance in both 1.0 M KOH and chlorine-alkali electrolyte are compared. As shown in Figure R6a, the overpotential at 10 mA cm⁻² for Cs₃Rh₂I₉/NC-R, Cs₃Rh₂I₉-R and Rh/NC are 25, 123 and 31 mV, respectively. The Tafel slope of Cs₃Rh₂I₉/NC-R is only 30.3 mV dec⁻¹, lower than that of Rh/NC (35.5 mV dec⁻¹). And Cs₃Rh₂I₉/NC-R showed large current density of 0.065 mA cm⁻²_{ECSA} at the overpotential of 50 mV, 30% higher than that of Rh/NC without GB, indicating its high intrinsic activity. In the simulated chlorine-alkali electrolyte, the overpotentials for Cs₃Rh₂I₉/NC-R are 21, 65, and 107 mV to reach current densities of 10, 50 and 100 mA cm⁻², respectively, significantly lower than those of Cs₃Rh₂I₉-R and Rh/NC (Figure R6c). The mass activity of Cs₃Rh₂I₉/NC-R is 782.8 mA mg⁻¹_{Rh} at -50 mV vs. RHE, outperforming Cs₃Rh₂I₉-R (22.1 mA mg⁻¹_{Rh}) and Rh/NC (175.9 mA mg⁻¹_{Rh}). These results confirm the excellent alkaline HER activity of Cs₃Rh₂I₉/NC-R.

Figure R6 HER activity in 1.0 M KOH and chlor-alkali electrolyte. **a** LSV curves of all samples in 1.0 M KOH. **b** Comparison of overpotential at 10 mA cm⁻² and Tafel slope for various Rh-based catalysts in 1.0 M KOH. **c** LSV curves of all samples in chlorine-alkali electrolyte. **d** Mass activity normalized to the mass of Rh/Pt. **e** Comparison of mass activity and area activity at the overpotential of 50 mV in chlorine-alkali electrolyte. **f** Stability of Cs₃Rh₂I₉/NC-R at 10 mA cm⁻² in chlorine-alkali electrolyte.

Through the comparison of current density and normalized activity for Cs₃Rh₂I₉/NC-R, Cs₃Rh₂I₉-R and Rh/NC, it is reasonable to exclude the influence of particle size and

substrate on HER performance and thereby confirming the role of GB in Cs₃Rh₂I₉/NC-R.

Moreover, the enhanced water dissociation ability in Cs₃Rh₂I₉/NC-R was verified by in situ Raman spectra. Figure R7a shows obvious interfacial water on Cs₃Rh₂I₉/NC-R surface in 1M KOH solution over the potential range from 0 to -0.3 V (vs. RHE). The peaks at 1588 and 1632 cm⁻¹ correspond to the G band of carbon substrate and the adsorbed water on Rh, respectively (Nature, 1965, 205, 170, J. Electroanal. Chem., 1996, 415, 175-178). As the potential decreased, the intensity of H-O-H bending increases sharply. However, such phenomenon is absent in Rh/NC (Figure R7b). The strong H-O-H bending sign on twined Rh with abundant grain boundaries suggest the high water dissociation kinetics on Cs₃Rh₂I₉/NC-R during electrocatalysis.

Figure R7 In situ Raman spectra of interfacial water on Cs₃Rh₂I₉/NC-R (a) and Rh/NC (b).

And the DFT calculation was also performed to confirm the role of GB and tensile stress in HER. In alkaline HER, the water dissociation, as a pre-reaction to form an adsorbed proton, plays a more critical role in electrocatalysis. As shown in Supplementary Fig. 39, the increase of H-O-H angle and water molecule binding energy agrees well with degree of distortion and crystal tension, indicating the enhanced ability of water dissociation. Therefore, GBs are proved to play the key role in enhanced the hydrogen evolution activity.

Reviewer #3 (Remarks to the Author):

The authors reported the Rh nanoparticles with abundant GBs, and detailly elucidated the synthetic process and analyzed the formed mechanism of GB. In particular, the top-down and bottom-up methods for Rh nanoparticles preparation were compared by many physical characterization techniques. As expected, the GB-enriched Rh nanoparticles demonstrated the excellent catalytic activity, which is superior to monocrystalline Rh without GBs. It claims that such activity results from the boosted water dissociation kinetics by the GB with strain stress. However, there are many critical issues (see following) that prevent me from recommending this paper to publish in the high-quality journal.

1. The authors claim that the GB-enriched Rh nanoparticles were prepared by the bottom-up methods based on the NC used as support. Although many reported literatures have similar description, namely, there is a substrate as a nucleation site that is considered as bottom-up methods. However, the specific process of electrochemical reduction of Cs₃Rh₂I₉ perovskite clusters is no basically comprehensive research.

Response:

Thanks very much for your positive comments. We have tried to observe the reduction process of Cs₃Rh₂I₉ perovskite clusters by TEM, whereas the clusters are unstable under the high-energy measurement (Figure R8). The average size of Cs₃Rh₂I₉ clusters is only 1.7 nm. However, the size of Rh are 2.2 nm after reduction due to the aggregation of smaller Rh clusters during the electrochemical reduction process. To elucidate the process of reconstruction, the reduction was conducted by the potentiostatic measurement at -0.03 V vs. RHE. The in-situ electrochemical Quartz Crystal Microbalance (EQCM) experiment, ex-situ Inductively Coupled Plasma Mass Spectrometry (ICP-MS) and XPS spectra characterizations were carried out to elucidate the Cs and I extraction and Rh reduction during the electrochemical reduction.

Figure R8 **a** HAADF-STEM of $\text{Cs}_3\text{Rh}_2\text{I}_9/\text{NC}$ and **b** decomposed $\text{Cs}_3\text{Rh}_2\text{I}_9/\text{NC}$.

The in-situ EQCM measurement indicates that the quality of the electrode continuously decreases and then stabilizes after about 5 min (Figure R9a). The ICP-MS results show the content of Cs^+ ions in the electrolyte continuously increases in the initial 5 min while Rh in the electrolyte was barely detected (Figure R9b). The ex-situ XPS spectra verify the content of Rh^{3+} decreases while the content of Rh^0 increases with the reduction time (Figure R9c). Therefore, the Cs and I gradually extracted from the perovskite clusters and Rh^{3+} was reduced to Rh^0 during the reconstruction. Meanwhile, the small Rh clusters combined with neighboring clusters into particles (~ 2.2 nm) to decrease the surface energy (Figure R10).

Figure R9 **a** Mass change of the $\text{Cs}_3\text{Rh}_2\text{I}_9/\text{NC}$ electrode monitored by in-situ EQCM experiment. **b** ICP-MS of the Cs and Rh contents in the electrolyte at different reduction time. **c** XPS spectra of Rh 3d for $\text{Cs}_3\text{Rh}_2\text{I}_9/\text{NC}$ at different reduction time.

Figure R10 HAADF-STEM images of adjacent Rh particles in reduced $\text{Cs}_3\text{Rh}_2\text{I}_9/\text{NC}$ ($\text{Cs}_3\text{Rh}_2\text{I}_9/\text{NC-R}$).

2. There is no experimental investigation to verify the enhanced water dissociation kinetics. The DFT calculations is only the theory.

Response:

Thanks very much for Reviewer's suggestion. In situ Raman was taken to prove the enhanced water dissociation kinetics of the catalyst. Figure R11a shows obvious interfacial water on $\text{Cs}_3\text{Rh}_2\text{I}_9/\text{NC-R}$ surface in 1M KOH solution over the potential range from 0 to -0.3 V (vs. RHE). The peaks at 1588 and 1632 cm^{-1} correspond to the G band of carbon substrate and the adsorbed water on Rh, respectively (Nature, 1965, 205, 170, J. Electroanal. Chem., 1996, 415, 175-178). As the potential decreased, the intensity of H-O-H bending increases sharply. However, such phenomenon is absent in Rh/NC (Figure R11b). The strong H-O-H bending sign on twined Rh with abundant grain boundaries suggest the high water dissociation kinetics on $\text{Cs}_3\text{Rh}_2\text{I}_9/\text{NC-R}$ during electrocatalysis. This result is consistent with the DFT calculations.

Figure R11 In situ Raman spectra of interfacial water on Cs₃Rh₂I₉/NC-R (a) and Rh/NC (b).

We have added Figure R11 as Supplementary Fig. 40 in Supplementary Information. The corresponding discussion was also added in the main text:

The enhanced water dissociation ability in Cs₃Rh₂I₉/NC-R was further verified by in situ Raman spectra. Supplementary Fig. 40a shows obvious interfacial water on Cs₃Rh₂I₉/NC-R surface in 1M KOH solution over the potential range from 0 to -0.3 V (vs. RHE). The peaks at 1588 and 1632 cm⁻¹ correspond to the G band of carbon substrate and the adsorbed water on Rh, respectively^{45, 46}. As the potential decreased, the intensity of H-O-H bending increases sharply. However, such phenomenon is absent in Rh/NC without GBs (Supplementary Fig. 40b).

3. According to the XPS and EDX with limited detection ability, it is not certain that there is no Cs and I elements in Rh nanoparticles. Thus, the HER activity origin needs to be further investigated because the synergistic effect between elements is very important in enhancing HER performance.

Response:

Thanks very much for Reviewer's suggestion. As the synergistic effect between elements is very important in enhancing HER performance, it is necessary to confirm the contents of Cs and I in Rh nanoparticles. ICP-MS results show the mass ratio of Rh, Cs and I is 1:0.0084:0.0024 in Cs₃Rh₂I₉/NC-R, indicating the product was reduced to Rh

particles. The small amount of Cs and I maybe ascribed to the adsorbed Cs and I ions on the NC substrate. The ICP-MS results have been added in the Supplementary Table 5.

4. In term of DFT calculations, especial the constructed atom model, authors did not consider the role of NC in HER process, but both XPS and XAS results evidenced the strong electronic interactions between Rh and NC.

Response:

As reviewer pointed, both XPS and XAS results evidenced the strong electronic interactions between twined Rh particles and NC substrate. As the electronic interactions may influence the catalytic performance, it is necessary to set up reference sample to rule out the role of NC in the HER process. Rh/NC prepared by chemical reduction of RhCl_3 with sodium borohydride was served as a benchmark. XPS results show binding energies of $\text{Rh}^0 3d_{3/2}$ and $\text{Rh}^0 3d_{5/2}$ for Rh/NC are similar to those for $\text{Cs}_3\text{Rh}_2\text{I}_9/\text{NC-R}$ (Figure R12), indicating the electronic state of Rh in Rh/NC and $\text{Cs}_3\text{Rh}_2\text{I}_9/\text{NC-R}$ are the same. However, the HER activity of $\text{Cs}_3\text{Rh}_2\text{I}_9/\text{NC-R}$ outperforms Rh/NC in both 1.0 M KOH and chlorine-alkali electrolyte due to the grain boundaries and tension stress in $\text{Cs}_3\text{Rh}_2\text{I}_9/\text{NC-R}$. Therefore, grain boundaries play the key role in catalytic performance. Figure R12 was added as Supplementary Fig. 22c in Supplementary Information.

Figure R12 XPS spectra of Rh 3d for $\text{Cs}_3\text{Rh}_2\text{I}_9/\text{NC}$, $\text{Cs}_3\text{Rh}_2\text{I}_9/\text{NC-R}$ and Rh/NC.

5. In experimental method, there is a mistake. The RuCl₃ should be substituted with RhCl₃ in “synthesis of Rh/NC”.

Response:

Thanks very much for Reviewer’s careful check of our manuscript. The mistake has been corrected in the revised main text.

REVIEWER COMMENTS

Reviewer #1 (Remarks to the Author):

In addition to providing many atomic-scale STEM images for confirmation, the authors have appropriately addressed all the comments we made during the first round of review. We believe that this study is another noteworthy piece of work demonstrating that the lattice defects and GBs are significantly active sites for electrocatalysis in general. In this respect, we recommend this manuscript for rapid publication in Nature Communications.

Reviewer #2 (Remarks to the Author):

I appreciate the authors' comprehensive response. I acknowledge that the Issue 2-6 are fully resolved, and I am satisfied with the explanation provided.

However, I still have some concerns with regards to Issue 1, 7 and 8.

As mentioned in my original revision I was not convinced that the improvement in performance stemmed from grain boundaries. To certain extent the additional data provided by the authors seem to reinforce my doubts about it.

The authors state "In 1.0 M KOH, the GB-rich Cs₃Rh₂I₉/NC-R exhibited high mass activity of 846 mA mg⁻¹Rh at the overpotential of 50 mV, which is about 22.3 times that of GB-free Cs₃Rh₂I₉-R." This observation is unsurprising given that:

- (a) Cs₃Rh₂I₉/NC-R is immobilised on high surface area support;
- (b) there is significant reduction in particle sized. This becomes even more apparent from Figure R5. The initially polycrystalline (as evident from diffuse SAED) material is converted into highly crystalline crystals with substantially smaller sizes.

Furthermore, the authors still unanswered my initial concern "From the Fig. S20 (now Fig. S21) it seems that unsupported Rh-particles (aka Cs₃Rh₂I₉-R) are quite large, furthermore, plenty of grain boundaries are visible which seems to defy the hypothesis that improved characteristic of supported materials are due to grain boundaries." If both materials have grain boundaries than why only highly crystalline, small nanoparticles on high surface area substrate show improved performance? The provided Fig. S25 seems to be quite consistent with this observation as well, i.e. disproving that the performance is due to grain boundaries.

With regards to Point 8, I commend the authors on including the data for Rh-NP made from RhCl₃ and immobilised on NC which I found quite helpful for better understanding of the presented work. Before I

will be able to comment on Figure R6d which seems to be key to confirming that the Cs₃Rh₂I₉/NC-R is a superior catalyst over Rh-NC I would like the authors to provide information (or point out if it is within the manuscript / SI as I could not find anything) how the Rh loading (in mg) was assessed. From what I can see based on XPS (at least) both Rh-NC and Cs₃Rh₂I₉/NC-R are pretty much the same Rh nanoparticles. So technically the key idea of the manuscript should be more concerned with demonstrating whether Cs₃Rh₂I₉ is superior precursor to RhCl₃.

In summary, as before it seems that the additional evidence provided by the authors is consistent with my previous observation that "the role of the substrate (which improved dispersability and high active site exposure) and/or size of nanoparticles" is the reason from improved performance. As before I believe the manuscript is very well written and deserve publication in Nature Comms after the matter with the origin of improved catalytic performance is resolved.

Reviewer #3 (Remarks to the Author):

Comments: In this study, GB-enriched Rh twin nanoparticles, a catalyst for water dissociation, were prepared by a bottom-up evolution route of electrochemically reducing Cs₃Rh₂I₉ halide-perovskite clusters. This paper has been revised, but this revision is not satisfactory. Many of our concerns have not been well solved. Therefore, I do not recommend the acceptance of this paper. This paper is not suitable for publication in Nature Commun. There are some other issues should continue to be addressed.

1. XPS spectra at line 145-148 indicate that the "a positive core level shift of about 0.5 eV for Cs₃Rh₂I₉/NC, indicating that the strong carrier effect of Cs₃Rh₂I₉ clusters on the NC" is inappropriate, because two peaks shift to the same extent. It may be that the standard peak position is not corrected or affected by NC. It could not be determined as the strong carrier effect of Cs₃Rh₂I₉ clusters on the NC.
2. The evolution process of Rh³⁺ to twinned Rh is detailed in line 173 to 185. How will the reaction process change for Cs⁺ and I⁻ ions and why the content of Cs⁺ increases in the initial 5 min while Rh in the electrolyte was barely detected (Fig. 3c)? Can the authors give corresponding characterization or conjecture about the evolution process and structural changes of Cs₃Rh₂I₉ during the reduction reaction?
3. The formation of twinned Rh nanoparticle with rich GBs may be due to the uniform dispersion of NC to Cs₃Rh₂I₉ and the in-situ reduction at its adsorption site, while in the absence of NC, the formed bulk Cs₃Rh₂I₉ is agglomerated and subsequently forms large size Rh nanoparticle upon reduction. NC has a significant effect on the reduction of Cs₃Rh₂I₉ to twinned Rh nanoparticle rich with GBs. This depends on whether Cs₃Rh₂I₉ is adsorbed by the NC porous material. It is recommended to test the specific surface area of NC, Cs₃Rh₂I₉/NC and Cs₃Rh₂I₉/NC-R.

4. In the process of synthesis of Cs₃Rh₂I₉/NC, how to remove the free Cs₃Rh₂I₉ particles that are not adsorbed by NC? Anti-solvent water precipitates Cs₃Rh₂I₉ dissolved in DMF, and ethanol can dissolve free Cs₃Rh₂I₉?

Reviewer #1 (Remarks to the Author):

In addition to providing many atomic-scale STEM images for confirmation, the authors have appropriately addressed all the comments we made during the first round of review. We believe that this study is another noteworthy piece of work demonstrating that the lattice defects and GBs are significantly active sites for electrocatalysis in general. In this respect, we recommend this manuscript for rapid publication in Nature Communications.

Response:

We deeply appreciate the reviewing efforts and positive comments of the reviewer, which encourages us for further exploration in this field.

Reviewer #2 (Remarks to the Author):

I appreciate the authors' comprehensive response. I acknowledge that the Issue 2-6 are fully resolved, and I am satisfied with the explanation provided.

However, I still have some concerns with regards to Issue 1, 7 and 8.

Response:

We are sorry to hear that our previous response did not satisfy the reviewer. We sincerely hope that our response to the comments from this reviewer as well as other reviewers below will satisfactorily address the remaining concerns.

1. As mentioned in my original revision I was not convinced that the improvement in performance stemmed from grain boundaries. To certain extent the additional data provided by the authors seem to reinforce my doubts about it.

The authors state "In 1.0 M KOH, the GB-rich Cs₃Rh₂I₉/NC-R exhibited high mass activity of 846 mA mg⁻¹Rh at the overpotential of 50 mV, which is about 22.3 times that of GB-free Cs₃Rh₂I₉-R." This observation is unsurprising given that:

- (a) Cs₃Rh₂I₉/NC-R is immobilised on high surface area support;
- (b) there is significant reduction in particle sized. This becomes even more apparent from Figure R5. The initially polycrystalline (as evident from diffuse SAED) material is converted into highly crystalline crystals with substantially smaller sizes.

Response:

We appreciate the Reviewer' efforts and positive comments to improve the manuscript. As Reviewer pointed, the high surface area support plays a positive role in reducing the size of loaded particles to enhance the catalytic activity of Cs₃Rh₂I₉/NC-R. To prove the role of GB, the new Rh/NC (from RhCl₃) with small size of 2.4 ± 0.4 nm and Rh content of 7.0 wt.% was synthesized by chemical reduction for comparison. The particle size and mass content of Rh in Rh/NC are similar to those of Cs₃Rh₂I₉/NC-R (Rh content of 5.8 wt.%; size of 2.2 ± 0.3 nm), its area activity and mass activity are much lower. The details will be discussed in Issue 3.

Therefore, the question goes back to why the Cs₃Rh₂I₉-R with larger particle size was used as the comparison? In this work, we took advantage of the dissolution-precipitation property for the new compound Cs₃Rh₂I₉, and developed a bottom-up evolution route to prepare twinned Rh nanoparticles with high-density and exposed grain boundaries (GBs) *via* in-situ electrochemical reduction process. To emphasize the advantages of synthesized Cs₃Rh₂I₉/NC-R, the GBs-free Cs₃Rh₂I₉-R was set as the comparison. In Cs₃Rh₂I₉/NC-R, the Rh particles show smaller size with abundant GBs, which are responsible for the high HER activity.

However, the small size caused by the carbon support has been widely reported and confirmed in previous literatures. Thus, we paid more attention to the formed GBs within ultra-small particles. As the GBs-free Cs₃Rh₂I₉-R cannot fully demonstrate the advantages of GBs in HER, the GBs-free Rh/NC with similar particle size to Cs₃Rh₂I₉/NC-R synthesized by chemical reduction of RhCl₃ on NC was used for comparison. The Rh particle size of Rh/NC is 2.4 nm, very similar to 2.2 nm of Cs₃Rh₂I₉/NC-R. Therefore, the Rh/NC can further confirm the role of GBs in Cs₃Rh₂I₉/NC-R.

2. Furthermore, the authors still unanswered my initial concern “From the Fig. S20 (now Fig. S21) it seems that unsupported Rh-particles (aka Cs₃Rh₂I₉-R) are quite large, furthermore, plenty of grain boundaries are visible which seems to defy the hypothesis that improved characteristic of supported materials are due to grain boundaries.” If both materials have grain boundaries than why only highly crystalline, small nanoparticles on high surface area substrate show improved performance? The provided Fig. S25 seems to be quite consistent with this observation as well, i.e. disproving that the performance is due to grain boundaries.

Response:

As shown in Fig. R1, the Cs₃Rh₂I₉-R is consisted of numerous Rh particles after electrochemical reduction. HRTEM images further show the dense accumulation of nano-crystals (Figure R1b and c). However, the abundant boundaries are the edges of

each particle, instead of twin GBs within the particles. In sharp contrast, Figure R2 (Fig. S25 in previous version) exhibits rich GBs within the nanoparticles for Cs₃Rh₂I₉/NC-R, which is different from that in Cs₃Rh₂I₉-R. Therefore, the Rh particles in Cs₃Rh₂I₉-R are single crystalline without GBs. To further highlight the key role of twin GBs in Rh, the Rh/NC without GBs was synthesized by chemical reduction of RhCl₃ on NC for comparison. Please see the next reply.

Figure R1 a TEM of Cs₃Rh₂I₉ after reduction (Cs₃Rh₂I₉-R). b-e HRTEM of Cs₃Rh₂I₉-

R. The inset in **c** shows particle size distribution.

Figure R2 HAADF-STEM images of $\text{Cs}_3\text{Rh}_2\text{I}_9/\text{NC-R}$.

3. With regards to Point 8, I commend the authors on including the data for Rh-NP made from RhCl_3 and immobilised on NC which I found quite helpful for better understanding of the presented work. Before I will be able to comment on Figure R6d which seems to be key to confirming that the $\text{Cs}_3\text{Rh}_2\text{I}_9/\text{NC-R}$ is a superior catalyst over Rh-NC I would like the authors to provide information (or point out if it is within the manuscript / SI as I could not find anything) how the Rh loading (in mg) was assessed. From what I can see based on XPS (at least) both Rh-NC and $\text{Cs}_3\text{Rh}_2\text{I}_9/\text{NC-R}$ are pretty much the same Rh nanoparticles. So technically the key idea of the manuscript should be more concerned with demonstrating whether $\text{Cs}_3\text{Rh}_2\text{I}_9$ is superior precursor to RhCl_3 .

Response:

The mass loading of Rh/NC in previous version is 20 wt.%, which were mentioned in the section of *Synthesis of Rh/NC without GBs* in *Method*. To better confirm the role of

GBs in enhanced HER activity, the Rh/NC (Rh content of 7.0 wt.%) with particle size of 2.4 ± 0.4 nm was synthesized to replace the previous Rh/NC (20 wt.% Rh content). The new Rh/NC (7 wt.% Rh content) shows similar particle size, mass loading as well as electronic structure to the $\text{Cs}_3\text{Rh}_2\text{I}_9/\text{NC-R}$ (Figure R3), and can be used as comparison to clarify the role of GBs in boosting the HER.

Figure R3 **a** TEM images of Rh/NC (Rh content: 7.0 wt.%). The inset shows the HRTEM of Rh particles. **b** Particle size distribution from **a**. **c** XPS spectra of Rh 3d for $\text{Cs}_3\text{Rh}_2\text{I}_9/\text{NC}$, $\text{Cs}_3\text{Rh}_2\text{I}_9/\text{NC-R}$ and Rh/NC.

In 1.0 M KOH, the overpotentials at 10 mA cm^{-2} for $\text{Cs}_3\text{Rh}_2\text{I}_9/\text{NC-R}$, $\text{Cs}_3\text{Rh}_2\text{I}_9\text{-R}$, and Rh/NC are 25, 123 and 41 mV (Figure R4a), respectively. $\text{Cs}_3\text{Rh}_2\text{I}_9/\text{NC-R}$ composed of Rh twin nanoparticles also exhibits higher mass activity than $\text{Cs}_3\text{Rh}_2\text{I}_9\text{-R}$ and Rh/NC. To exclude the influence of particle size, the activity was also normalized to ECSA. The area activity of $\text{Cs}_3\text{Rh}_2\text{I}_9/\text{NC-R}$ is $0.065 \text{ mA cm}^{-2}\text{ECSA}$, which is obvious higher than the $0.029 \text{ mA cm}^{-2}\text{ECSA}$ for $\text{Cs}_3\text{Rh}_2\text{I}_9\text{-R}$ and $0.040 \text{ mA cm}^{-2}\text{ECSA}$ for Rh/NC (Figure R4b).

In simulated chlorine-alkali electrolyte, the overpotentials for Cs₃Rh₂I₉/NC-R are 21, 65, and 107 mV to reach current densities of 10, 50 and 100 mA cm⁻² (Figure R5a), respectively, significantly lower than those of Cs₃Rh₂I₉-R and Rh/NC. The mass activity of Cs₃Rh₂I₉/NC-R is 782.8 mA mg⁻¹_{Rh} at -50 mV vs. RHE, outperforming Cs₃Rh₂I₉-R (22.1 mA mg⁻¹_{Rh}) and Rh/NC (305.4 mA mg⁻¹_{Rh}) (Figure R5b). Moreover, the area activity of Cs₃Rh₂I₉/NC-R manifests a factor of 1.1 increase than that of Rh/NC without GBs (Figure R5c), indicating its high intrinsic activity.

By the comparison of current density and normalized activity for Cs₃Rh₂I₉/NC-R, Cs₃Rh₂I₉-R and Rh/NC, it is reasonable to exclude the influence of particle size and substrate on HER performance, thereby confirming the role of GBs in Cs₃Rh₂I₉/NC-R toward the HER.

Figure R4 HER activity in 1.0 M KOH. **a** LSV curves of various electrocatalysts including commercial Pt/C (Pt content: 20 wt.%) coated on rotating GCE (1600 rpm) in 1.0 M KOH. The catalyst loading amount on GCE is 0.764 mg cm⁻². For Rh-based samples, the calculated Rh loading amounts on GCE are 0.045, 0.090 and 0.053 mg cm⁻² for Cs₃Rh₂I₉/NC-R, Cs₃Rh₂I₉-R, and Rh/NC, respectively. **b** Comparison of mass activity and area activity of Cs₃Rh₂I₉/NC-R, Cs₃Rh₂I₉-R, and Rh/NC at the overpotential of 50 mV.

Figure R5 HER activity in chlorine-alkali electrolyte. a LSV curves of various electrocatalysts including commercial Pt/C (Pt content: 20 wt.%) coated on rotating GCE (1600 rpm) in chlorine-alkali electrolyte. The catalyst loading amount on GCE is 0.764 mg cm^{-2} . For Rh-based samples, the calculated Rh loading amounts on GCE are 0.045, 0.090 and 0.053 mg cm^{-2} for $\text{Cs}_3\text{Rh}_2\text{I}_9/\text{NC-R}$, $\text{Cs}_3\text{Rh}_2\text{I}_9\text{-R}$, and Rh/NC, respectively. **b** Mass activity normalized to the mass of Rh. **c** Area activity normalized to the ECSA. **d** Comparison of mass activity and area activity at the overpotential of 50 mV.

We have added Figure R3 as new Supplementary Fig. 26 in Supplementary Information. The electrochemical data of new Rh/NC (7 wt.% Rh content) have updated in main text and Supplementary Information.

4. In summary, as before it seems that the additional evidence provided by the authors is consistent with my previous observation that “the role of the substrate (which improved dispersability and high active site exposure) and/or size of nanoparticles” is the reason from improved performance. As before I believe the manuscript is very well

written and deserve publication in Nature Comms after the matter with the origin of improved catalytic performance is resolved.

Response:

Thank you very much for your patience and valuable comments. We believe the new synthesized Rh/NC with Rh content of 7 wt.% and particle size of ~2.4 nm can be served as the suitable comparison to prove the role of GBs in enhanced HER activity. And in the section of *HER activity evolution* and *Discussion*, we have emphasized the comparison of area activity of Cs₃Rh₂I₉/NC-R and Rh/NC to clarify the higher intrinsic activity of GB-enriched Rh. We sincerely hope that our above response will satisfactorily address the remaining concerns.

Reviewer #3 (Remarks to the Author):

Comments: In this study, GB-enriched Rh twin nanoparticles, a catalyst for water dissociation, were prepared by a bottom-up evolution route of electrochemically reducing Cs₃Rh₂I₉ halide-perovskite clusters. This paper has been revised, but this revision is not satisfactory. Many of our concerns have not been well solved. Therefore, I do not recommend the acceptance of this paper. This paper is not suitable for publication in Nature Commun. There are some other issues should continue to be addressed.

Response:

We are sorry to hear that our previous response did not satisfy the reviewer. We sincerely hope that our response to the comments from this reviewer below will satisfactorily address the remaining concerns.

In this work, polycrystalline Rh nanoparticles on NC with size of ~2.2 nm and exposed grain boundaries were synthesized by the *in-situ* electrochemical reduction of Cs₃Rh₂I₉/NC. The new compound Cs₃Rh₂I₉ perovskite with unique zero-dimensional structure was developed and nanosized as precursor for the polycrystalline Rh. Various *in-situ* and *ex-situ* characterizations including electrochemical quartz crystal microbalance experiments were carried out to elucidate the Cs and I extraction and Rh reduction during the electrochemical reduction. And polycrystalline Rh nanoparticles on NC were formed by aggregation of Rh clusters driven by the surface energy. Such a bottom-up process promotes the high-density grain boundaries within Rh nanoparticles, which also induce tensile stress into the nearby Rh atoms. In 1.0 M KOH, the catalyst Cs₃Rh₂I₉/NC-R showed a low overpotential of 25 mV at the current density of 10 mA cm⁻² and a small Tafel slope of 30.3 mV dec⁻¹. In chlor-alkali electrolyte, its area activity (0.069 mA cm⁻²_{ECSA} at -50 mV vs. RHE) manifests a factor of 3.2 and 1.1 activity increase compared to GB-free monocrystalline Rh and Rh/NC, respectively. Our work not only designs the polycrystalline nano-catalyst with high catalytic activity, but also offers a fresh insight on the reaction mechanism of grain boundary and tensile atoms.

1. XPS spectra at line 145-148 indicate that the "a positive core level shift of about 0.5 eV for Cs₃Rh₂I₉/NC, indicating that the strong carrier effect of Cs₃Rh₂I₉ clusters on the NC" is inappropriate, because two peaks shift to the same extent. It may be that the standard peak position is not corrected or affected by NC. It could not be determined as the strong carrier effect of Cs₃Rh₂I₉ clusters on the NC.

Response:

Thank you very much for carefully checking our manuscript.

All XPS spectra in this manuscript were calibrated by the C 1s. XPS measurement is a very useful technique to determine the chemical states of materials. For example, the negative shift of the binding energy for Ru⁰ 3d_{5/2} and Ru⁴⁺ 3d_{5/2} in Ru/TiO₂ confirms the electronic interaction between Ru and TiO₂ support (Nat. Catal., 2020, 3, 454-462); the shift of the Pd 3d_{5/2} and Pd 3d_{3/2} XPS peaks of PdMo bimetallic to higher binding energy relative to those of Pd metallic indicates the electronic effect of Mo doping (Nature, 2019, 574, 81-85). Therefore, the positive core level shift of Rh 3d for Cs₃Rh₂I₉/NC indicates the strong carrier effect of Cs₃Rh₂I₉ clusters on the NC. This phenomenon also occurs on Pt/NC in our previous report (Adv. Energy Mater., 2021, 11, 2101050). The binding energy of Pt 4f shows the 0.4 eV shift when loading on the substrate.

2. The evolution process of Rh³⁺ to twinned Rh is detailed in line 173 to 185. How will the reaction process change for Cs⁺ and I⁻ ions and why the content of Cs⁺ increases in the initial 5 min while Rh in the electrolyte was barely detected (Fig. 3c)? Can the authors give corresponding characterization or conjecture about the evolution process and structural changes of Cs₃Rh₂I₉ during the reduction reaction?

Response:

Thank you very much for the comments. As shown in Figure R6a, the reduction peak at about -0.035 V versus reversible hydrogen electrode (*vs.* RHE) in the first CV curve corresponds to the reduction of Rh³⁺. The peak was only observed in the first cycle, suggesting the reduction process proceeds quickly. To elucidate the process of the

electrochemical reduction, the reduction was carried out by potentiostatic measurement at -0.03 V vs. RHE , and the in-situ electrochemical Quartz Crystal Microbalance (EQCM) experiment. The in-situ EQCM experiment can provide the mass change of the electrode during the reduction. Figure R6b shows that the quality of the electrode continuously decreases and then stabilizes after about 5 min, indicating that a part of the catalyst is dissolved and detached from the electrode during reduction process, and such phenomenon lasted for 5 minutes. Therefore, ICP-MS was used to evaluate the ions content in the electrolyte to identify dissolved elements. Figure R6c shows the Rh didn't dissolve and Cs dissolved into the electrolyte. The ex-situ XPS spectra verify the content of Rh^{3+} decreases while the content of Rh^0 increases with the reduction time (Figure R6d), confirming the Rh^0 remains on the surface of NC.

The HAADF-STEM images indicate the Rh nanoparticles in $\text{Cs}_3\text{Rh}_2\text{I}_9/\text{NC-R}$ are rich with twin-GBs and larger plane spacing (Fig. 4e in main text). In sharp contrast, the particle in reduced $\text{Cs}_3\text{Rh}_2\text{I}_9$ ($\text{Cs}_3\text{Rh}_2\text{I}_9\text{-R}$) is monocrystalline (Fig. 4f in main text). The formation of twinned Rh from $\text{Cs}_3\text{Rh}_2\text{I}_9/\text{NC}$ may be ascribed to the coupling of smaller Rh clusters with high surface energy during the electrochemical reduction process. Limited by the size of the $\text{Cs}_3\text{Rh}_2\text{I}_9$ cluster ($\sim 1.7\text{ nm}$), the formed Rh clusters could combine with neighboring clusters into particles ($\sim 2.2\text{ nm}$) to decrease the surface energy.

Figure R6 **a** The CV curves from 1st to 100th cycle at 100 mV s⁻¹ for Cs₃Rh₂I₉/NC in 1.0 M KOH. **b** Mass change of the Cs₃Rh₂I₉/NC electrode monitored by in-situ EQCM experiment. **c** ICP-MS of the Cs and Rh contents in the electrolyte at different reduction time. **d** XPS spectra of Rh 3d for Cs₃Rh₂I₉/NC at different reduction time. The results in panel b-d were obtained at the potentiostatic measurement at -0.03 V vs. RHE.

3. The formation of twinned Rh nanoparticle with rich GBs may be due to the uniform dispersion of NC to Cs₃Rh₂I₉ and the in-situ reduction at its adsorption site, while in the absence of NC, the formed bulk Cs₃Rh₂I₉ is agglomerated and subsequently forms large size Rh nanoparticle upon reduction. NC has a significant effect on the reduction of Cs₃Rh₂I₉ to twinned Rh nanoparticle rich with GBs. This depends on whether Cs₃Rh₂I₉ is adsorbed by the NC porous material. It is recommended to test the specific surface area of NC, Cs₃Rh₂I₉/NC and Cs₃Rh₂I₉/NC-R.

Response:

The nitrogen adsorption-desorption isotherms indicate the increased adsorption volume at low relative pressure for Cs₃Rh₂I₉/NC relative to that of NC (Figure R7), suggesting the increased micropore surface area after loading the Cs₃Rh₂I₉ perovskite clusters. As Reviewer pointed, the NC has a significant effect on the reduction of Cs₃Rh₂I₉ to twined Rh particles rich with GBs due to the support effect. Besides, the NC can stabilize the Cs₃Rh₂I₉ with the size of 1.7 ± 0.2 nm by strong carrier effect, proved by the XPS spectra.

Figure R7 Nitrogen adsorption-desorption isotherms.

4. In the process of synthesis of Cs₃Rh₂I₉/NC, how to remove the free Cs₃Rh₂I₉ particles that are not adsorbed by NC? Anti-solvent water precipitates Cs₃Rh₂I₉ dissolved in DMF, and ethanol can dissolve free Cs₃Rh₂I₉?

Response:

In this work, the polar nitrogen-doped carbon (NC) can be served as nucleation sites for the dissolved Cs₃Rh₂I₉. When the mixture of NC and dissolved Cs₃Rh₂I₉ were slowly added in the H₂O, the Cs₃Rh₂I₉ particles were preferentially precipitated on the surface of NC. As shown in Figure R8, no obvious free Cs₃Rh₂I₉ are observed, indicating that most of particles with very small size are loaded on the NC. In sharp contrast, the HRTEM confirm the formation of ultra-small sized Cs₃Rh₂I₉ nanoclusters on the NC (Fig. 2f in main text)

Figure R8 SEM images of Cs₃Rh₂I₉/NC.

The polar aprotic solvents, including dimethylformamide (DMF), dimethyl sulfoxide (DMSO), dimethylacetamide (DMAC) and N-methyl-2-pyrrolidone (NMP) are commonly used as solvents for halide perovskite (Nat. Sustain. 2021, 4, 277-285 and ACS Energy Lett. 2019, 4, 2983-2985). However, the solubility of halide perovskite in polar protic solvent (H₂O and ethanol) is much lower due to the slow S_N2 reaction (Nat. Commun. 2016, 7, 11735 and Rev. Chem. Soc. 1962, 16, 163). Thus, the Cs₃Rh₂I₉ clusters could be loaded on the NC when the mixture of NC and dissolved Cs₃Rh₂I₉ were slowly added in the H₂O. Moreover, the Cs₃Rh₂I₉ can't dissolve in ethanol.

We have added Figure R8 as new Supplementary Fig. 9 in Supplementary Information.

REVIEWER COMMENTS

Reviewer #2 (Remarks to the Author):

1. As before the comparison between Figure R1 and R2 seems to indicate that Cs₃Rh₂I₉-R and Cs₃Rh₂I₉/NC-R are both contain some grain boundaries. For example, the particle in Fig. R1e appear pretty much the same as in Fig. R2a. The assessment whether Cs₃Rh₂I₉-R have grain boundaries is purely through the eye of beholder and better images are needed to make it. Therefore, I would appreciate if authors would provide HAADF-STEM images for Cs₃Rh₂I₉-R.

2. Similar situation seems to persist when authors are comparing between two types of Rh nanoparticles: (1) Cs₃Rh₂I₉/NC-R and (2) Rh/NC. While the authors tried hard to find some grain boundaries for Cs₃Rh₂I₉/NC-R the images in Fig. R3 are obscure. As above, HAADF-STEM images are required for Rh/NC to allow for comparison with Cs₃Rh₂I₉/NC-R sample. This should hopefully close the matter as it will provide like for like comparison for the readers.

3. I appreciate the authors preparing and testing the new sample with 7 wt. % loading of Rh. The elemental analysis was carried out which makes it quite well characterised sample in terms of Rh-content. Has there been any attempts to evaluate the actual amount of Rh loading in Cs₃Rh₂I₉/NC-R? I understand that sample is generated in situ but it should be possible to scrap off some of it from the RDE as elemental analysis requires fairly small amounts.

4. Now, if we assume that Cs₃Rh₂I₉/NC-R and Rh/NC (7 wt. %) have similar loadings from LSV their performance is practically identical (within the error of the measurements). The difference is quite marginal and although Cs₃Rh₂I₉/NC-R is somewhat better it would be a very difficult to attribute the minor improvement to grain boundaries. There might be a range of different factors that affect the performance of the samples. The biggest factor is sample preparation on RDE as the authors drop cast catalyst on the electrode. Therefore, the variation between the OVP from different samples would be sufficient to account for difference in performance (as samples would tend to dry differently upon immobilization on RDE).

5. I wonder why the authors obtained different capacitance values (ECSA) for Cs₃Rh₂I₉/NC-R and Rh/NC (7 wt. %). If the authors used the same carbon support than the capacitance values would stem mostly from NC and give similar ECSA for Cs₃Rh₂I₉/NC-R and Rh/NC (7 wt. %). The difference in capacitance implies that drop casted samples adhere somewhat differently to the electrodes causing the difference in performance. In other words, the difference between two samples prepared by two different methods is not necessarily due to grain boundaries.

Overall, I still think it is a good work, showing a new method of immobilisation of Rh-nanoparticles on carbon support which seemed to lead to some improvement in catalytic activity. As there are numerous factors at play, I still can't see how grain boundaries are solely responsible for (minor) difference in performance. Further evidences are required or the authors may just describe what their data show without trying to overemphasize on grain boundaries in Cs₃Rh₂I₉/NC-R. This could be achieved by changing the narrative a bit with a prime focus on presenting Cs₃Rh₂I₉ as a viable alternative to RhCl₃.

Reviewer #2 (Remarks to the Author):

Response:

We appreciate the Reviewer' efforts and positive comments to improve the manuscript. We have revised the manuscript according to your valuable comments. In the main text, the role of grain boundary has softened. The positive effects of tensile stress, metal atom coordination state, enlarged lattice spacing, carbon substrate and accessible electrochemical surface area are also mentioned. And liquid-synthesized Rh/NC is set as the comparison to emphasize the advantages of the new halide-perovskite compound $\text{Cs}_3\text{Rh}_2\text{I}_9$ as well as the in-situ complete reconstruction. Detailed discussions are listed in the following point-by-point response. Hopefully the revised edition could satisfy you.

1. As before the comparison between Figure R1 and R2 seems to indicate that $\text{Cs}_3\text{Rh}_2\text{I}_9\text{-R}$ and $\text{Cs}_3\text{Rh}_2\text{I}_9/\text{NC-R}$ are both contain some grain boundaries. For example, the particle in Fig. R1e appear pretty much the same as in Fig. R2a. The assessment whether $\text{Cs}_3\text{Rh}_2\text{I}_9\text{-R}$ have grain boundaries is purely through the eye of beholder and better images are needed to make it. Therefore, I would appreciate if authors would provide HAADF-STEM images for $\text{Cs}_3\text{Rh}_2\text{I}_9\text{-R}$.

Response:

Thank you for your suggestions. The edge of $\text{Cs}_3\text{Rh}_2\text{I}_9\text{-R}$ is consisted of numerous Rh particles after electrochemical reduction (Figure R1). The densely packing of nanoparticles make it hard to clarify whether the boundary is a particle edge or a grain boundary inside particles. Therefore, more HRTEM images are provided to prove most particles in $\text{Cs}_3\text{Rh}_2\text{I}_9\text{-R}$ do not have grain boundary.

Figure R1 HRTEM of Cs₃Rh₂I₉-R. The inset in **b** shows particle size distribution.

We have added Figure R1 as new Supplementary Fig. 21 in Supplementary Information.

2. Similar situation seems to persist when authors are comparing between two types of Rh nanoparticles: (1) Cs₃Rh₂I₉/NC-R and (2) Rh/NC. While the authors tried hard to find some grain boundaries for Cs₃Rh₂I₉/NC-R the images in Fig. R3 are obscure. As above, HAADF-STEM images are required for Rh/NC to allow for comparison with Cs₃Rh₂I₉/NC-R sample. This should hopefully close the matter as it will provide like for like comparison for the readers.

Response:

Thank you very much for your kind comments. More HRTEM images are provided in Figure R2. According to Reviewer's suggesting, we have weakened the statement of grain boundary to prevent propagating the erroneous perception to the readers.

Figure R2 **a** TEM images of Rh/NC. The inset shows the Rh particle size distribution. **b** and **c** HRTEM of Rh/NC. The red circles show the Rh particle.

We have added Figure R2 as new Supplementary Fig. 26 in Supplementary Information.

3. I appreciate the authors preparing and testing the new sample with 7 wt. % loading of Rh. The elemental analysis was carried out which makes it quite well characterised sample in terms of Rh-content. Has there been any attempts to evaluate the actual amount of Rh loading in Cs₃Rh₂I₉/NC-R? I understand that sample is generated in situ but it should be possible to scrap off some of it from the RDE as elemental analysis requires fairly small amounts.

Response:

Thank you very much for your nice comments. The Cs₃Rh₂I₉/NC-R on the RDE was collected by the ultrasound in ethanol solution. Its Rh mass content was determined by the ICP-MS to be 5.7 ± 0.8 wt.%, which is very close to the initial loading of 5.8 wt.%.

4. Now, if we assume that Cs₃Rh₂I₉/NC-R and Rh/NC (7 wt.%) have similar loadings from LSV their performance is practically identical (within the error of the measurements). The difference is quite marginal and although Cs₃Rh₂I₉/NC-R is somewhat better it would be a very difficult to attribute the minor improvement to grain boundaries. There might be a range of different factors that affect the performance of the samples. The biggest factor is sample preparation on RDE as the authors drop cast

catalyst on the electrode. Therefore, the variation between the OVP from different samples would be sufficient to account for difference in performance (as samples would tend to dry differently upon immobilization on RDE).

Response:

Thank you for your kind suggestions. The homogeneous catalyst ink for RDE was prepared by mixing 5 mg catalyst and 25 μL Nafion (5 wt.%) with 475 μL ethanol. After ultrasonic treatment for 30 min, 15 μL ink was dropped on a rotating GCE (area: 0.1963 cm^2). It was subsequently dried in an air-dry oven at $50\text{ }^\circ\text{C}$ to form the uniform catalyst film. In 1.0 M KOH, the current densities at the overpotential of 50 mV for $\text{Cs}_3\text{Rh}_2\text{I}_9/\text{NC-R}$, $\text{Cs}_3\text{Rh}_2\text{I}_9\text{-R}$, and Rh/NC are 37.3 ± 0.6 , 3.4 ± 0.1 and $18.8 \pm 0.3\text{ mA cm}^{-2}$, respectively (Figure R3). In chlor-alkali electrolyte, the current densities at the overpotential of 50 mV for $\text{Cs}_3\text{Rh}_2\text{I}_9/\text{NC-R}$, $\text{Cs}_3\text{Rh}_2\text{I}_9\text{-R}$, and Rh/NC are 33.7 ± 0.7 , 1.9 ± 0.2 and $16.5 \pm 0.8\text{ mA cm}^{-2}$, respectively (Figure R4). Above results show the error of the measurements is negligible. Therefore, the enhanced activity of $\text{Cs}_3\text{Rh}_2\text{I}_9/\text{NC-R}$ compared to Rh/NC is attributed to its unique structure.

Figure R3 The HER activity of different electrocatalysts on rotating GCE (1600 rpm)

in 1.0 M KOH. **a-c** Three independent LSV curves for $\text{Cs}_3\text{Rh}_2\text{I}_9/\text{NC-R}$, $\text{Cs}_3\text{Rh}_2\text{I}_9\text{-R}$, and Rh/NC. **d** Comparison of mass activity and area activity at the overpotential of 50 mV. All data shows the mean and standard deviation through three repeated measurements.

Figure R4 The HER activity of different electrocatalysts on rotating GCE (1600 rpm) in chlor-alkali electrolyte. **a-c** Three independent LSV curves for $\text{Cs}_3\text{Rh}_2\text{I}_9/\text{NC-R}$, $\text{Cs}_3\text{Rh}_2\text{I}_9\text{-R}$, and Rh/NC. **d** Comparison of mass activity and area activity at the overpotential of 50 mV. All data shows the mean and standard deviation through three repeated measurements.

We have added Figure R3d and Figure R4d as new Fig. 5e (in main text) and Supplementary Fig. 28d (in Supplementary Information), respectively.

5. I wonder why the authors obtained different capacitance values (ECSA) for $\text{Cs}_3\text{Rh}_2\text{I}_9/\text{NC-R}$ and Rh/NC (7 wt. %). If the authors used the same carbon support

than the capacitance values would stem mostly from NC and give similar ECSA for Cs₃Rh₂I₉/NC-R and Rh/NC (7 wt. %). The difference in capacitance implies that drop casted samples adhere somewhat differently to the electrodes causing the difference in performance. In other words, the difference between two samples prepared by two different methods is not necessarily due to grain boundaries.

Response:

Thanks for Reviewer's careful check of our manuscript. The double layer capacitance is a quantitative indicator of the surface area that is accessible to the electrolyte ions (J. Am. Chem. Soc., 2015, 137, 4347-4357; J. Am. Chem. Soc., 2013, 135, 16977-16987). Metal particle with large surface also exhibits obvious double-layer charging in non-Faradaic potential range (Chemical Society Reviews, 2019, 48, 2518-2534; J. Am. Chem. Soc., 2018, 140, 2397-2400; Electrochimica Acta, 2001, 46, 3063-3071). In this work, the Rh nanoparticles with ultrasmall size contributes to the most double layer capacitance due to the ECSA increased with the mass loading (Figure R5).

Figure R5 The electrochemical surface area (ECSA) for various samples.

6. Overall, I still think it is a good work, showing a new method of immobilisation of Rh-nanoparticles on carbon support which seemed to lead to some improvement in catalytic activity. As there are numerous factors at play, I still can't see how grain boundaries are solely responsible for (minor) difference in performance. Further

evidences are required or the authors may just describe what their data show without trying to overemphasize on grain boundaries in Cs₃Rh₂I₉/NC-R. This could be achieved by changing the narrative a bit with a prime focus on presenting Cs₃Rh₂I₉ as a viable alternative to RhCl₃.

Response:

Thank you very much for your patience and valuable comments. As Reviewer pointed, it is hard to prove grain boundaries are solely responsible for enhanced performance. Tensile stress, metal atom coordination state, enlarged lattice spacing, carbon substrate and accessible electrochemical surface area also can lead to the difference of catalytic activity. Therefore, we have revised the statement tone of the main text and focus on presenting the reconstruction process of Cs₃Rh₂I₉/NC. The title was change from “*Bottom-up Evolution of Perovskite Clusters into High-activity Rhodium Nanoparticles Enriched with Grain Boundary*” to “*Bottom-up Evolution of Perovskite Clusters into High-activity Rhodium Nanoparticles toward Alkaline Hydrogen Evolution*”. In ABSTRACT, we paid more attention on presenting Cs₃Rh₂I₉ as a viable alternative to RhCl₃ to obtain the small size Rh nanoparticles with high alkaline hydrogen evolution in both 1.0 M KOH and chlor-alkali electrolyte. Moreover, the corresponding part in INTRODUCTION and RESULTS were also modified. The revisions have been highlighted by yellow in the revised text. We sincerely hope that our response to the comments from this reviewer will satisfactorily address the remaining concerns.

REVIEWERS' COMMENTS

Reviewer #2 (Remarks to the Author):

I am delighted with the improved version of the manuscript and the narration focused on a catalyst performance. I accept all changes and wish the authors all the best with their manuscript and future work.